

1. Aircraft-engine particulate matter emissions from conventional

2. and sustainable aviation fuel combustion: comparison of

3. measurement techniques for mass, number, and size

4. Joel C. Corbin[a], Tobias Schripp[b], Bruce E. Anderson[c], Greg J.

5. Smallwood[a], Patrick LeClercq[b], Ewan C. Crosbie[c,d], Steven Achterberg[e],

6. Philip D. Whitefield[e], Richard C. Miake-Lye[f], Zhenhong Yu[f], Andrew

7. Freedman[f], Max Trueblood[e], David Satterfield[e], Wenyan Liu[e], Patrick

8. Oßwald[b], Claire Robinson[c,d], Michael A. Shook[c], Richard H. Moore[c] and

9. Prem Lobo[a]

10. [a]Metrology Research Centre, National Research Council Canada, Ottawa, Ontario,

11. Canada

12. [b]German Aerospace Center (DLR), Institute of Combustion Technology, Stuttgart,

13. Germany

14. [c]NASA Langley Research Center, Hampton, Virginia, USA

15. [d]Science Systems and Applications, Inc., Hampton Virginia, USA

16. [e]Department of Chemistry, Missouri University of Science and Technology, Rolla,

17. Missouri, USA

18. [f]Aerodyne Research, Inc., Billerica, Massachusetts, USA

19. Correspondence to: Joel C. Corbin(Joel.Corbin@nrc-cnrc.gc.ca) and Prem Lobo

20. (Prem.Lobo@nrc-cnrc.gc.ca)



0    AMT Feature: short summary (max. 500 characters incl. spaces)

The combustion of sustainable aviation fuels in aircraft engines produces particulate matter (PM) emissions with different properties than conventional fuels due to changes in fuel composition. Consequently, the response of various diagnostic instruments to PM emissions may be impacted. We found no significant instrument biases in terms of particle mass, number, and size measurements for conventional and sustainable aviation fuel blends despite large differences in the magnitude of emissions.

# 1    Abstract

Sustainable aviation fuels (SAFs) have different compositions compared to conventional petroleum jet fuels, particularly in terms of fuel sulphur and hydrocarbon content. These differences may change the amount and physicochemical properties of volatile and non-volatile particulate matter (nvPM) emitted by aircraft engines. In this study, we evaluate whether comparable nvPM measurement techniques respond similarly to nvPM produced by three blends of SAFs compared to three conventional fuels. Multiple SAF blends and conventional (Jet A-1) jet fuels were combusted in a V2527-A5 engine, while an additional conventional fuel (JP-8) was combusted in a CFM56-2C1 engine.

We evaluated nvPM mass concentration measured by three real-time sampling techniques: photoacoustic spectroscopy, laser-induced incandescence, and the extinction-minus-scattering technique. Various commercial instruments were tested including three LII 300s, one PAX, one MSS+, and two CAPS $PM_{SSA}$. Mass-based emission indices ($EI_m$) reported by these techniques were similar, falling within 30% of their geometric mean for $EI_m$ above 100 $mg/kg_{fuel}$ (approximately 10 μg PM $m^{-3}$ at the instrument), this geometric mean was therefore used as a reference value. Additionally, two integrative measurement techniques were evaluated: filter photometry and particle size distribution (PSD) integration. The commercial instruments used were one TAP, one PSAP, and two SMPSs. These techniques are used in specific applications, such as on-board research aircraft to determine PM emissions at cruise. $EI_m$ reported by the alternative techniques fell within approximately 50 % of the mean aerosol-phase $EI_m$.





In addition, we measured PM-number-based emissions indices using PSDs and
condensation particle counters. The commercial instruments used included TSI
SMPSs, a Cambustion DMS500, and an AVL APC, and the data also fell within
approximately 50 % of their geometric mean. The number-based emission indices
were highly sensitive to the accuracy of the sampling-line penetration functions
applied as corrections. In contrast, the $EI_m$ data were less sensitive to those
corrections since a smaller volume fraction fell within the size range where
corrections were substantial. A separate, dedicated experiment also showed that
the operating laser fluence used in the LII 300 laser-induced incandescence
instrument for aircraft engine nvPM measurement is adequate for a range of SAF
blends investigated in this study. Overall, we conclude that all tested instruments
are suitable for the measurement of nvPM emissions from the combustion of SAF
blends in aircraft engines.
Keywords: non-volatile particulate matter, aircraft, emissions, sustainable
aviation fuels, black carbon
2   Introduction
Aircraft engine particulate matter (PM) emissions are composed of non-volatile
(black carbon, metal ash, oxygenated functional groups) and volatile components
(volatile organic compounds, nitrates, sulphates) (Gagné et al., 2021; Masiol and
Harrison, 2014; Petzold et al., 2011). The non-volatile particulate matter (nvPM)
emissions are formed in the combustor, while volatile particulate matter (vPM)
emissions, present in the gas phase at the engine exit, condense after emission.
Aircraft engines emit vPM with similar or greater orders of magnitude as nvPM,
especially after the vapour pressure of volatile species is lowered by oxidative
aging (Kiliç et al., 2018) or by cooling (Beyersdorf et al., 2014). The nvPM and vPM
are constituents of total PM which affects air quality, health, and climate. The
International Civil Aviation Organization (ICAO) has developed standards and
recommended practices (SARPs) for measuring the mass- and number-based
emissions of nvPM emitted from aircraft engines with maximum rated thrust >26.7
kN (ICAO, 2017). Currently, SARPs have not been established for vPM or total PM
(Lobo et al., 2020). The SARPs for nvPM specify standardized sampling and



measurement protocols (SAE, 2013, 2018; ICAO, 2017), which have been
extensively evaluated and validated (Lobo et al., 2015b, 2020; Kinsey et al., 2021).
The nvPM regulatory limits are applicable for type certification of aircraft engines,
but they do not address the vPM which may have substantial environmental
impacts.

To reduce $CO_2$ emissions, mitigate environmental impacts, and make the aviation
sector more sustainable, a significant effort is underway to develop and deploy
sustainable aviation fuels (SAFs). Various feedstocks and different conversion
pathways can be used to produce SAFs (Hileman and Stratton, 2014), which differ
in chemical and physical properties compared to conventional petroleum jet fuel
(Vozka et al., 2019), most notably by lacking aromatic and sulfur species that are
precursors to nvPM and vPM emissions. New SAF candidates must undergo a
rigorous qualification and approval process (ASTM D4054) prior to being certified
under the ASTM D7566 standard specification as a blending component. Currently,
the ASTM D7566 standard allows SAF blend ratios of up to 50% with conventional
fuel for drop-in fuels (Wilson et al., 2013).

The combustion of neat SAFs and blends with conventional jet fuel has been shown
to result in different PM emissions characteristics as a function of engine type and
operating condition (Beyersdorf et al., 2014; Brem et al., 2015; Corporan et al.,
2011; Lobo et al., 2011, 2015a, 2016; Moore et al., 2017; Schripp et al., 2018, 2019;
Timko et al., 2010). In addition to changes in PM mass- and number-based
emissions, SAF combustion results in changes to particle size distributions (PSD)
(Beyersdorf et al., 2014; Cain et al., 2013; Kinsey et al., 2012; Lobo et al., 2011,
2015a, 2016; Schripp et al., 2018; Timko et al., 2010), chemical composition (Elser
et al., 2019; Kinsey et al., 2012; Timko et al., 2013; Williams et al., 2012),
morphology (Huang and Vander Wal, 2013; Kumal et al., 2020; Liati et al., 2019),
hygroscopic properties (Trueblood et al., 2018), and optical properties (Elser et al.,

112  2019).


The standardized sampling and measurement protocol for aircraft engine nvPM
emissions was designed and validated for engine certification tests using





conventional jet fuel. The SARP requires that number-based nvPM emissions are
measured with a butanol-based condensation-nuclei counter with 10 nm 50% cut-
size sampling in single-particle-counting mode downstream of a diluter and
catalytic stripper.  For mass-based nvPM emissions, the instrument must be
insensitive to vPM and able to meet performance specifications for repeatability,
zero drift, linearity, limit of detection, rise time, sampling interval, accuracy, and
applicability.   To date, the only commercial instruments that satisfy the SARP
number and mass measurement system requirements, respectively, are the AVL
Particle Counter (APC) Advanced, and the AVL Micro Soot Sensor (MSS) and the
Artium Laser Induced Incandescence LII 300 instrument (LII). Limited information
is available on aircraft engine nvPM emissions characteristics measured with the
standardized system for different engine types burning SAFs and blends with
conventional fuel (Durand et al., 2021; Elser et al., 2019; Lobo et al., 2015a, 2016).

The standardized system components are not easily adaptable for use on aircraft
for measurement of cruise level nvPM emissions.  Consequently, there are no
comparable in-flight engine-emissions data available for developing and validating
models that predict cruise nvPM-emissions based on engine certification data.
Particle size distribution measurements are also not included in the standardized
system, which are important for assessing the effects of fuels, operating conditions,
and engine technologies on the environmental impacts of PM emissions. Thus to
advance our understanding of aircraft engine emissions and the factors that
control them as well as to develop a large and consistent observational data base,
it is important to evaluate the relative performance of other diagnostic
instruments that are not prescribed in the standardized protocol but meet these
needs. Such instruments must be evaluated for their response to nvPM and total
PM emissions from aircraft engines using standardized and non-standardized
systems, and for measurements at the engine exit plane and downstream of the
engine in the near field, since these instruments are typically used with minimal
change to their operating parameters for a wide range of sampling conditions.




The observations presented in this paper were collected during the NASA/DLR-
Multidisciplinary Airborne Experiment (ND-MAX)/ Emission and Climate Impact
of Alternative Fuel (ECLIF) 2 campaign that was conducted at Ramstein Air Base,
Ramstein-Miesenbach, Germany in January-February 2018 (see overview by
(Anderson and NDMAX-Team, 2021)). The campaign included ground-based and
in-flight measurements of emissions from the DLR Advanced Technology Research
Aircraft (ATRA) A320 aircraft with V2527-A5 engines running on two
conventional jet fuels and three blends with SAF. The main objective of the ground-
based measurements was to characterize the nvPM, total PM, and hydrocarbon
emissions as functions of engine thrust condition and fuel composition. Several
identical instruments were included in the in-flight sampling aircraft (NASA DC-8)
and ground measurement diagnostic instrument suites to enable comparisons of
engine emissions during ground and airborne operations, and create a data set for
testing cruise emission models. The NASA DC-8 aircraft with CFM56-2C1 engines
was also used as an emissions source to compare various emissions diagnostic
instruments during the ground-based measurements.

Here we present the inter-comparison of real-time measurements of aircraft
engine nvPM emissions in terms of physical characteristics such as mass, number,
and size distributions using different diagnostic instruments and measurement
principles. The nvPM mass emissions were evaluated using three real-time
sampling techniques: photoacoustic spectroscopy, the extinction-minus-scattering
technique, and laser-induced incandescence (LII), and two alternative
measurement techniques widely used in laboratories and on-board aircraft: filter-
based photometry and PSD integration. We note that one of the photoacoustic
instruments and the LII instruments have been demonstrated to be compliant with
the ICAO SARP performance specifications. The PM number-based emissions were
measured using a condensation particle counter. The PSD characteristics
measured by scanning mobility particle sizers and an electrical mobility
spectrometer were also compared. The nvPM and total PM emissions were
delineated using a thermal denuder and a catalytic stripper.  We also report the
effect of laser fluence on the laser-induced incandescence of nvPM for SAF
combustion as changing carbon nanostructure is known to influence particle light





absorption and consequently LII signals, and hence the derived nvPM mass
concentration. The impact of fuel composition on PM emissions will be reported
separately (Schripp and NDMAX-Team, 2021).

## 3  Methods

### 3.1  Engine and fuels

In the majority of this work, emissions were sampled from a single IAE
V2527-A5 starboard engine of the DLR ATRA aircraft (Airbus A320-232). The
engine was operated on two conventional, petroleum jet fuels, referred to as REF3
and REF4, and three sustainable aviation fuel blends, referred to as SAF1, SAF2,
and SAF3. The abbreviations for the two conventional petroleum fuels are used to
avoid confusion with the previous ECLIF campaign (Schripp et al., 2018).
A limited number of experiments were also performed with  JP-8 fuel,
combusted in the starboard CFM56-2C1 engine (#3) of the NASA DC-8 aircraft.
Due to limited fuel availability, none of the other five fuels could be combusted in
the CFM56-2C1 engine. The properties of the six fuels are summarized in Table 1.

### 3.2  Ambient conditions

The measurements presented in this manuscript were performed outdoors
during winter in western Germany. Detailed meteorology for each test point is
given in the supplement. The minimum, median, and maximum temperatures were
2.3, 2.9, 8.3 °C, respectively. Conditions were humid (>83 % humidity) and
sometimes rainy. Winds ranged from 0 to 15.5 km h$^{-1}$ and wind direction was
sometimes variable. The median wind direction was south-westerly, while the
source aircraft was oriented facing to the east. Consequently, winds blowing
approximately 45$^{o}$ angle from the right rear of the source aircraft sometimes
prevented the engine emissions from reaching the sampling probe at low engine
thrust settings.



### 3.3 Emissions sampling

An extensive suite of aerosol and gas-phase instruments operated by the members of six different institutions were deployed in two different shipping containers to characterize the emissions (Table 2). The complete emission-sampling setup is discussed in companion papers (Anderson and NDMAX-Team, 2021; Schripp and NDMAX-Team, 2021). Briefly, emissions were sampled through a probe located 43 m downstream of the starboard engine of the aircraft. The probe was placed in front of a blast fence located on the western side of the Ramstein Air Force Base flight line, and the fence redirected the engine exhaust upwards for safety. The probe was connected to a 18.5-mm ID, 20-m-long electrically-conductive sampling line heated to 60 °C, that transported flow to a sampling plenum maintained at 33 °C. To minimize residence time and particle losses in this sampling line, a pump ensured that a total of at least 137 L min$^{-1}$ flowed through the sampling manifold at all times. Higher flows produce an unacceptably large pressure drop in the primary sampling line. The majority of this flow was discarded as excess.

The plenum was placed inside a modified shipping container (Container 1) behind the blast fence, along with the NRC, DLR, and NASA instruments. The North American Reference System (NARS) was connected to the plenum by a short section of heated line to the NARS dilutor box, which was heated to 60 ± 15 °C and contained a custom Dekati dilutor with a dilution ratio of approximately 4 (less than the standard Dekati dilutor ratio of 8 to 14). A 25 m line heated to 60 ± 15 °C transferred sample aerosols flow from the dilutor box to a second shipping container (Container 2), where the MST and ARI instruments were connected in parallel. The NARS components include the 25 m heated line, attached diluters and MST instrument suite; the system is compliant with specifications for the standardized nvPM sampling and measurement system (SAE, 2013; SAE, 2018; ICAO, 2017) and whose performance has been demonstrated and evaluated in previous studies (Lobo et al., 2015b, 2016, 2020). Additional instrumentation installed as part of the NARS included a fast electrical mobility spectrometer (Cambustion DMS500), an Aerodyne Aerosol Mass Spectrometer (results not





presented here), and a CAPS PMssa monitor (Aerodyne Research Inc.). The details
of the instruments installed inside these two containers are listed in Table 3.

### 3.3.1 Gaseous measurements

A suite of gaseous emissions was measured in this study, as summarized in Table
2. The $CO_2$ measurements from the NASA LI-COR 7000 were in good agreement
with those taken by DLR (MKS MultiGas 2030 FTIR Continuous Gas Analyzer) and
MST (LI-COR model 840A), but had a faster response time and were therefore used
as the reference for instruments in Container 1. Instruments in Container 2 used
the MST measurements as reference.

### 3.3.2 nvPM number and particle size distributions (PSDs)

nvPM number concentration was measured directly by a certification-test-
compliant, particle counter, APC (AVL Inc., which contains a TSI Model 3790E
CPC), which was part of the NARS in Container 2. PSDs were measured with two
technologies: scanning mobility particle sizers (SMPS, TSI Inc.) and electrical
mobility sizers (EMS). Two types of EMS were used; the Cambustion DMS500 (in
Container 2, measuring particles 10 to 1000 nm in diameter) and the TSI Engine
Exhaust Particle Sizer (EEPS, Container 1). However, the EEPS data were excluded
from this analysis due to unidentified problems with the instrument which led to
anomalous PSDs.
Two SMPSs measured nvPM PSDs. An SMPS operated by NRC measured
particles 10 to 278 nm in diameter downstream of a catalytic stripper (Model
CS015, Catalytic Instruments GmbH), which heated samples to 350 °C before
oxidizing gas-phase VOCs to prevent them from recondensing after exiting the
device. Another SMPS operated by NASA measured particles 10 to 278 nm in
diameter either directly or downstream of a NASA-constructed thermal denuder
(TD) also operated at 350 °C. The TD employs a concentric activated charcoal filter
downstream of the sample heater to prevent re-condensation of volatile species.
TDs are commonly used on-board aircraft for measuring nvPM number
concentration and size distributions (Clarke, 1991; Moore et al., 2017) and have
been shown to effectively evaporate nucleation and accumulation mode sulfate
and organic aerosols (Beyersdorf et al., 2014; Schripp et al., 2018).



### 3.3.3 nvPM mass measurements

In this study, most of the nvPM mass data were derived from light absorption coefficients (units of $m^{-1}$), either determined in flow-through sample cells (the CAPS $PM_{SSA}$, PAX, and MSS introduced below) or after collecting particles onto a filter (the TAP and PSAP introduced below). Such absorption measurements can be converted to equivalent black carbon or eBC mass concentrations (eBC, units of $g\,m^{-3}$; Petzold et al. (2013)) by dividing them by a reference mass absorption cross-section (MAC, units of $m^2\,g^{-1}$). The LII measurements also rely on light absorption, although the measurand is not absorption but incandescence at two wavelengths and is termed rBC (Petzold et al., 2013; Michelsen et al., 2014).

The reference MAC used to report eBC represents an assumed physical property of the nvPM emitted by the engine at a given time. The extensive review of Bond and Bergstrom (2006) concluded that the MAC at 550 nm of externally-mixed BC from a variety of sources could be summarized as $7.5 \pm 1.2\ m^2\,g^{-1}$; the more recent review of in-situ measurements by (Liu et al., 2020) recommended $8.0 \pm 0.7\ m^2\,g^{-1}$ at 550 nm. In this study, we have used the Bond and Bergstrom value of $7.5\ m^2\,g^{-1}$ for consistency with earlier work and instrument software. These values are assumed to vary inversely with wavelength, with an Angstrom (power) exponent of 1; for example, the 660 nm CAPS $PM_{SSA}$ monitor data were processed with a MAC of $7.5\ m^2\,g^{-1} \times (550\ nm\ /\ 660\ nm)^1 = 6.5\ m^2\,g^{-1}$.

One eBC technique, the CAPS $PM_{SSA}$ monitor (Aerodyne Research Inc.; Onasch et al., 2015) derives absorption coefficients as the difference between measured aerosol extinction and scattering coefficients, from which eBC concentrations were calculated as described above. The CAPS $PM_{SSA}$ measures light extinction by the calibration-free cavity attenuation phase shift (CAPS) technique and light scattering with an integrating nephelometer. The CAPS technique measures the lifetime of photons in a high-finesse optical cavity comprised of two high reflectivity mirrors, from which the extinction coefficient can be calculated. An integrating nephelometer captures light scattered from a section of this cavity, and is calibrated using the measured extinction of small (Rayleigh regime) non-absorbing particles. In this study, two CAPS $PM_{SSA}$ were present, one operated at 630 nm wavelength by ARI and the other at 660 nm wavelength by NRC. The



scattering channel of the NRC CAPS PM$_{SSA}$ was calibrated on-site using nebulized
and dried ammonium sulfate particles; the other instruments were similarly
calibrated prior to the campaign at the manufacturer using 200 nm ammonium
sulfate. For the sub-200 nm particles measured in this study, no truncation
corrections (Modini et al., 2021) were necessary.
Two other eBC instruments were based on photoacoustic spectroscopy, namely
the Photoacoustic Extinctiometer (PAX, DMT Inc,; Nakayama et al., 2015) and the
Micro Soot Sensor (MSS; AVL GmbH; Schindler et al., 2004). In both of these
instruments, aerosol absorption is measured by the periodic heating of particles
using a modulated laser, resulting in the generation of pressure waves which are
amplified by an acoustic cell and detected by a microphone. The PAX was
calibrated using nebulized ammonium sulfate as well as graphitic nanoparticles
(Aquadag).
During on-site calibration of the PAX using graphitic Aquadag nanoparticles, the
PAX signals were observed to drifted slowly upwards after each baseline. We were
nevertheless able to obtain useful data by configuring the PAX to auto-baseline
every 180 seconds, and only using the first 15 seconds of measurements after each
baseline. After the campaign, it was found that a component of the circuit board
was damaged during the initial shipment. In spite of this electrical problem, the
PAX data do not represent outliers in the following analysis.
Two additional pairs of eBC instruments were deployed at the ground site and on-
board the NASA DC-8 that measured aerosol absorption coefficients based on filter
attenuation, namely a Particle Soot Absorption Photometer (PSAP, Radiance
Research; Bond et al., 1999) and Tricolor Absorption Photometer (TAP, Brechtel
Manufacturing Inc, ; Ogren et al., 2017). These instruments were designed as low-
cost, low-maintenance devices for monitoring aerosol optical properties in the
background atmosphere (i.e., at low concentrations) and have been used
previously in airborne and ground-based studies (Moore et al., 2017). In these
instruments, particles are continuously collected onto an internal filter while its
light attenuation is measured. The change in light attenuation over time is used to
calculate absorption coefficients. This calculation requires post-processing to





correct for filter loading effects (which do not require independent measurements)
and may also be corrected for light attenuation due to scattering rather than
absorption (which requires an independent nephelometer measurement)
(Virkkula, 2010). Other sources of error include nonlinearities due to size-
dependent penetration of particles into the filter media and the evaporation of
volatile species over time (Lack et al., 2014; Nakayama et al., 2010). We note that
the TAP automatically advances its filter when its transmission drops below 80%,
whereas the PSAP requires a manual filter change. The PSAP filter was therefore
changed manually before each set of experiments herein, to ensure that its filter
transmission remained above 80% during all measurements.
Finally, three Artium LII 300 (Artium Technologies) instruments measured rBC,
based on two-colour pulsed laser–induced incandescence (LII) (Snelling et al.,
2005). These instruments heat nvPM using a 1064 nm pulsed laser and measure
the resulting incandescence at two wavelength bands. From this measurement,
rBC temperature and mass concentrations can be calculated. One of the LII 300s
was a component of the NARS. Of the other two, one was dedicated to an
experiment where its operating conditions were varied (Section 4.6). Therefore,
only two LII 300s were measuring real-time nvPM mass concentration
simultaneously at any given time. The MSS+ and the LII 300s were calibrated by
reference to the elemental carbon mass (defined by thermal–optical analysis)
produced by a laboratory diffusion-flame combustion aerosol source using
measurements at three mass concentrations spanning 0.1 to 0.5 mg m$^{-3}$ (SAE,

356 2018).

### 3.4 Data analysis

#### 3.4.1 Emission index calculations

The raw data were analysed over comparable time intervals and cross-
checked by independent calculations. The general analysis proceeded as described
in this section. First, the time series of measured $CO_2$ concentrations was used as a
reference against which to synchronize all time series, based on rapid rises and
falls in the observed concentrations (measured at 1 Hz) when the engine thrust





condition underwent large changes (as shown at 08:00 in Figure 2). All
instruments were synchronized against the NASA $CO_2$ sensor except the
instruments in container 2, which was synchronized against the MST LI-COR $CO_2$
sensor, because of the additional dilution stage. The time synchronization
accounted for different lag times due to differences in the response times and clock
accuracy of each instrument.
Second, the $CO_2$ concentrations [$CO_2$] were baseline-subtracted and filtered as
follows. The CO2 baseline ([$CO_2$]$_b$) was calculated as the mean of the $CO_2$
concentrations measured before ([$CO_2$]$_0$) and after ([$CO_2$]$_1$) each test. The
uncertainty in this baseline value was calculated as either ([$CO_2$]$_b$ – [$CO_2$]$_0$) or
([$CO_2$]$_b$ – [$CO_2$]$_1$), whichever was greater.
Due to the prevailing crosswind mentioned above, unstable $CO_2$ concentrations
occurred during from some test points at the idle engine thrust condition. These
unstable conditions were identified and filtered using two separate methods. In
the first method, the SMPS PSDs were inspected for reproducibility. In the second
method, an algorithm was used to reject any test points with $CO_2$ uncertainties
greater than 50%, $CO_2$ signals less than a factor of ten greater than uncertainty, or
$CO_2$ signals less than 20% above baseline. We found that the first method rejected
all of the points rejected by the algorithm, in addition to a few additional points.
The analysis presented uses the first method.
Third, all data were arithmetically averaged over the test point periods defined in
Table S1. For each instrument, the averaging periods were refined by inspection of
the data since sampling-line residence times varied. The averaged data were
typically at 1 Hz sampling frequency initially, although the SMPS instruments
measured PSDs at 45 second intervals (NRC instrument) or 30 second intervals
(NASA). Emission indices (EIs) were then calculated from the averaged data
following (SAE, 2013):

$$EI_m = PM_m \frac{RT_m}{[CO_2](M_c + \alpha M_H)P_m} \tag{1}$$






$$EI_{num} = PN \times 10^6 \frac{RT_m}{[CO_2](M_c + \alpha M_H)P_m} \tag{2}$$


Where $EI_m$ and $EI_{num}$ are mass and number-based EIs, respectively; $PM_m$
and PN are mass and number concentrations, respectively, at standard reference
temperature ($T_m$; 273.15 K) and pressure ($P_m$; 1 atm); $\alpha$ is the hydrogen to carbon
ratio of the fuel; $M_c$ and $M_H$ are the molar masses of carbon and hydrogen,
respectively; and $R$ is the ideal gas constant (0.082 L.atm.K$^{-1}$.mol$^{-1}$).
3.4.2    Loss correction
Particles may be lost to the walls of sampling lines or to deposits on those
walls. The fraction of particles penetrating a given system varies with size,
according to a characteristic penetration function. Four penetration functions were
applied in this study: 1) from the probe to the sampling plenum, 2) from the
plenum to the NARS, 3) within the TD, and 4) within the CS (Figure 4). Function 1
was measured on site as described below. Function 2 was calculated using the
standard loss calculation methodologies provided in SAE documents AIR6504
(SAE, 2017) and ARP6481 (SAE, 2019). Function 3 was experimentally determined
in the laboratory by NASA. Function 4 was obtained from theoretical estimates and
experimental measurements (Catalytic Stripper manual, 2014).

Penetration function 1 (probe-to-plenum penetration) was measured
experimentally by nebulizing ammonium sulfate particles at the probe while all
instruments were sampling and all heated lines had reached thermal equilibrium.
(Function 1 therefore also includes the smaller instrument sampling lines
downstream of the plenum in its correction as well; however, these were
considered negligible relative to the longer probe-to-plenum and plenum-to-
Container-2 transport lengths.) For this measurement, the NRC SMPS was moved
to the probe, while the NASA instrument remained in its standard position. The
ratio of the NASA to NRC PSDs then provided a first estimate of the penetration
function. However, this first estimate was not accurate, as the measurements were
performed on a cold day (measured as approximately 5 °C outdoors and 15 °C in
the instrument container) and as it does not account for performance differences


between the NASA and NRC SMPSs. Therefore, two corrections were made. First,
both measurements were corrected to standard temperature and pressure.
Second, differences between the two instruments were directly measured by
moving the NRC SMPS just outside of the sampling container (to keep it at 5 °C)
and connecting it to an identical sampling line as the NASA SMPS. The ratio of the
two measured PSDs in this setup was defined as equal to unity at all sizes, and
used to correct the initial penetration function. Therefore, no further correction
was made for sampling lines in Container 1. Losses in this additional line were
negligible (calculated penetrations of 0.997 at 100 nm and 0.98 at 20 nm) relative
to the long NARS line to Container 2 (i.e., Function 2).
3.5    Uncertainties

433        All reported uncertainties and error bars represent standard errors,

propagated through the calculation as necessary. When two independent sources
of uncertainty were available (for example, the standard error in the 10 second
averages of $[CO_2]$ and the uncertainty in the baseline value) they were added in
quadrature. Our bottom-up calculations of uncertainty can be compared with the
spread of the data points in our EI comparisons below. This spread represents a
top-down uncertainty, and is similar in magnitude to the bottom-up uncertainties
(i.e. error bars). This similarity lends confidence to our uncertainty estimates. In
most figures, error bars have generally been omitted for clarity, but uncertainties
are given for each instrument at each test point in Table S1.
4    Results and discussion
4.1    Experiment overview
A typical time series obtained when the emissions from the IAE V2527-A5 engine
were sampled is shown in Figure 2. Nominal low-pressure turbine fan speeds (N1)
expressed as a percentage of maximum continuous thrust, are shown by the labels
at the top of the figure. Percent N1 (along with engine fuel flow rate) is another
metric for representing the different engine thrust conditions and is used as a
primary independent variable in this study. The $CO_2$ concentrations (red line) were
highly variable at N1 = 23% as the ambient wind shifted the aircraft exhaust plume



toward and away from the sampling probe. Correspondingly, both nvPM mass and
PSD measurements were highly variable, as shown by the blue trace and black
symbols, respectively.

As shown in Figure 3a, nvPM mass concentrations, represented by $EI_m$, increased
with increasing N1 before decreasing slightly at the highest N1, similar to the
trends for other engine types reported by Lobo et al. (2015b, 2020). Figure 3b
shows that the relationship for $EI_{num}$ is less clear, with a slight increase at
moderate N1 followed by a greater decrease at high N1. As discussed below
(Section 4.2.3), the higher $EI_m$ at higher N1 thrust was associated with larger
particle sizes, and therefore smaller penetration-function corrections (Section
4.2.1). An effect of fuel composition is evident, and discussed in detail in Schripp et
al. (Schripp and NDMAX-Team, 2021).
## 4.2 Size distributions and penetration functions
### 4.2.1 Penetration function
A typical PSD, and corresponding PVD, are shown in Figure 4, in the context of the
penetration functions applied in this work. The PVD was calculated by assuming
spherical particles, which incurs negligible error for aircraft-engine nvPM due to
the small diameter of particles produced by such engines (Durdina et al., 2014;
Saffaripour et al., 2020). For the example PSD and PVD in Figure 4 (shading), it is
clear that a substantial fraction of the particle number was corrected for
penetrations (lines) of roughly 0.5. In contrast, the larger mode of the PVD
corresponds to penetrations larger than 0.8 in most cases. These differences led to
a median number- and mass-based correction factors of 1.51 and 1.19, respectively
for penetration Function 1 (probe to plenum) labelled in the figure. The remaining
instrument-specific penetration corrections were applied according to the position
of each instrument in the sampling system, as specified in Table 2. The magnitude
of each correction is given in Table S1.
Figure 5 shows selected PSDs from the IAE V2527-A5 engine operated with SAJF1
(Figure 5a) and REF4 (Figure 5b) fuels. The plot illustrates a lower (40 %) and a





higher thrust point (60 %) from the available data for two fuels. Note that the
ordinate scales are harmonized across the upper and lower rows only. All abscissa
scales are harmonized. The figure indicates roughly comparable PSDs from these
two fuels. The companion paper (Schripp and NDMAX-Team, 2021) compares the
effects of fuel composition in detail.
4.2.2   PSDs of CFM56-2C1
The CFM56-2C1 engine on the DC-8 burning JP-8 emitted an order of magnitude
more total particles per unit fuel burned than any of the fuels combusted in the
ATRA. We attribute this difference to the relatively high sulfur content of the JP-8
fuel (1490 ppm sulfur versus ≤ 105 ppm for the other fuels).  The CFM56-2C1
engine also emitted a factor of three lower nvPM mass and nvPM number than the
V2527-A5 engine. The presence of extremely small particles with $d_m < 10$nm was
evident in the two nvPM PSDs (not shown due to the extremely large penetration
function at these sizes; Figure 4). The CS-SMPS data extended to smaller diameters,
and showed that the size range measured by these two instruments was
insufficient to capture the full PSD for the CFM56-2C1 engine data at 22% N1 as
well as 63% N1.  The $d_m < 10$nm mode was not as prominent in the V2527-A5
engine exhaust at any thrust, although some evidence was observed for it (e.g.
number distribution at 40% N1 in Figure 5b).
Since the CFM56-2C1-with-JP-8 data were strongly influenced by a nucleation
mode, and were therefore not well described by the GMD and GSD of the data,
these measurements have been omitted from all subsequent PSD analysis in this
manuscript. Bimodal fits to the data were not possible as the nucleation mode was
not captured by our size distributions. However, the nvPM mass  measurements
are much less sensitive to these small particles (Hinds, 1999) and have therefore
been retained. PSDs from all instruments, test points, and fuels from both the
CFM56-2C1 and V2527-A5 engines are included in the supplement.
4.2.3   Particle size statistics; GMD and GSD
Figure 6 summarizes the PSDs measured by three instruments in terms of their





GMD and GSD. The data sets labelled SMPS and TD-SMPS were both obtained from
NASA's SMPS, which was manually switched to a bypass line and the TD at each
test point. The data set labelled CS-SMPS was obtained with NRC's SMPS.
Total PM is represented by the data sets labelled DMS500 and SMPS. However, the
two are not directly comparable because the DMS500 measurements were
obtained after an additional dilution by a factor of 4 in the NARS and the DMS500
was not operated behind a volatile particle remover (CS or TD). Moreover, the
inversion of DMS500 data requires more assumptions about the particle size
distribution than the analogous SMPS calculation. Either volatiles or this inversion
procedure may have caused the 10% larger GSDs observed for the DMS500 for
some data (some measurements with GMDs over 35 nm) relative to the SMPS.
Since volatiles would affect both GMD and GSD, but we primarily observed
discrepancies in the DMS500 GSD, we suggest that the inversion was the major
source of bias in these data.

nvPM is represented by the open circles and filled squares in Figure 6. These two
data sets show a different relationship (slope) between GMD and GSD, reflecting
systematic differences in the corresponding PSDs. Relative to the mean of the two
instruments, the NRC GMDs were higher (Figure 7a) while the NRC GSDs were
higher at GSD < 1.75 but lower at GSD > 1.75 (Figure 7b). Inspection of the
corresponding PSDs showed that the NASA and NRC instruments agreed at higher
$d_m$ but that NRC number concentrations were higher at smaller $d_m$. This trend
suggests that a bias in the penetration functions applied to each instrument
(Figure 4, Table 2) led to the discrepancy in GMD and GSD. Such a bias would affect
the nvPM concentration estimated from these PSDs (Figure 8b) and will be
discussed further below.
In spite of these trends in GMD and GSD, the PSD measurements agreed to within
20% (Figure 7a) for nvPM GMDs and within 5% for nvPM GSDs (Figure 7b).
Furthermore, these measurements are consistent with previous measurements by
Lobo et al. (2015c), as illustrated by the line in Figure 6, which reproduces the
polynomial best-fit line reported by those authors.



### 4.3 Consistency between number-based emission indices of nvPM and vPM

Figure 7c compares the measured vPM and nvPM $EI_{num}$ with the mean nvPM $EI_{num}$ (i.e., mean of the NRC CS-SMPS, NASA TD-SMPS, and NARS APC. The grey shading shows that all instruments agreed to within a factor of 2. The APC and DMS500 nvPM $EI_{num}$ were both typically higher than the two similar SMPSs. The APC has a 50% efficiency at its cut-off diameter of 10 nm, reaching 100% efficiency above this size and 0% below it. Therefore, relative to the SMPSs, which measured down to approximately 10 nm with 100% efficiency, the APC should measure lower than the SMPSs since it will underperform at sizes close to 10 nm. (This expectation requires that there are no particles present above the SMPS upper detection limit of 280 nm in our study, which was verified by our PSD analysis in Section 3 and Table 2). However, the APC measured approximately 50% larger nvPM $EI_{num}$ under all conditions, and our measured PSDs rule out the possibility that 50% of particles were not seen by the SMPS. Therefore, we attribute the difference between APC and SMPS results to uncertainties in the APC or SMPS penetration correction functions. Since the two SMPSs agreed, the APC measurements were likely overcorrected when the SARP correction procedures were applied.

We also attribute the larger nvPM $EI_{num}$ measured by the DMS500 to the same cause; to which a similar penetration function as the APC applies (Section 3.4.2).

### 4.4 Consistency between mass-based emission indices

#### 4.4.1 $EI_m$ measurements by real-time sampling instruments

Figure 8a presents scatterplots of the real-time $EI_m$ measurements acquired during this study for all fuels and both engine types. In Figure 8a, the individual $EI_m$ are plotted against the geometric mean of the instruments shown in the caption: three LII 300 instruments, two CAPS instruments, one PAX and one MSS+. The geometric mean was chosen over the more-common arithmetic mean because the data are not normally distributed; the arithmetic mean would therefore have over-emphasized outliers.





Figure 9a presents the same data as Figure 8a except that the measurements have
been normalized to the geometric-mean $EI_m$ from Figure 8a. Most data fall within
30 % of the mean (inner dashed lines) above 100 mg / $kg_{fuel}$. We note that exhaust
samples were diluted with background air by a factor of 40 or more before
reaching the inlet probe, so at this lower limit, the actual concentration observed
by the instruments was approximately 10 µg m$^{-3}$ (the exact conversion factor
varies with $CO_2$ concentration and fuel properties), which is close to their
detection limits, as expected. This lower limit may have been influenced by the
ambient measurement conditions, where background nvPM concentrations were
non-negligible.
The agreement of the real-time measurements to within 30 % is notable
considering the different types of instruments used. The scatter at lower EIm
values reflects the noise levels of the instruments. Both of these observations are
consistent with data reported previously for different engine types by Lobo et al.
(2016, 2020). The LII 300 and MSS+ from the North American Reference System
(NARS) have been widely used to characterize aircraft engine nvPM emissions. The
two CAPS instruments were independently calibrated and operated. The MSS+ and
PAX represent two photoacoustic spectrometers from different manufacturers,
operated by different teams, with different principles of calibration. The PAX was
also operated with a damaged capacitor on its printed circuit board. As noted in
Methods, these instruments operate on a variety of physical principles, including
photoacoustic spectroscopy (with two different designs), extinction-minus-
scattering, and laser-induced incandescence (cf. Section 3.3.3). Agreement
between these various principles also suggests that factors such as volatile
coatings on nvPM did not influence the instrument responses.
4.4.2   SMPS-based $EI_m$
Figure 8b and Figure 9b are analogous to Figure 8a and Figure 9a, but for the
integrative nvPM measurements that do not fall into the real-time sampling
category.  These data are plotted against the same geometric mean from Figure 8a.
The dashed lines in Figure 9b represent the same ratios as in Figure 9a.
Considering that the real-time instruments in Figure 8a were either calibrated to





aerosol absorption or to aviation nvPM, we consider their accuracy as greater than
the instruments in Figure 8b and consider departures from the 1:1 line as due to
inaccuracy.
Most of the instruments in Figure 8b were accurate to within 30% of the reference,
similar to Figure 8a, with the exception of the CS-SMPS and PSAP. This is
summarized in Table 3, which shows the mean ratios of all data except engine idle
(23% N1) with the geometric mean. Table 3 also includes the results of a linear
regression against the geometric mean to facilitate comparison of our
measurements with Kinsey et al. (2021), who performed linear regressions against
simultaneous elemental carbon (EC) measurements (in our study, mass
concentrations were too low to obtain EC measurements). The PSAP data are
discussed in the next section. The CS-SMPS data were systematically higher than
the geometric mean, potentially due to an overcorrection of the penetration of
large particles to the SMPS.
Since the spread of nvPM $EI_m$ reported by the two SMPS systems was smaller than
the bias, their difference relative to the reference $EI_m$ cannot be attributed to
measurement imprecision. Since the two SMPS systems showed different
accuracies, their differences cannot be ascribed to a lack of constraints on the
effective density of the nvPM particles (Momenimovahed and Olfert, 2015), which
may vary with the monomer diameter (Abegglen et al., 2014; Durdina et al., 2014)
and/or shape of soot aggregates. With respect to the real-time measurements, the
TD-SMPS data are also consistent with previous measurements of aviation engine
PSDs, which, however, were not corrected for diffusional particle loss (Lobo et al.,
2015b, 2020). Careful measurement of the penetration functions used in these
calculations would be required to confirm our interpretation.

### 4.4.3 Filter photometer-based $EI_m$ from TAP and PSAP

Figure 8b and Figure 9b show that the TAP measurements were within the 30 %
range observed for the real-time instruments, with a relative standard deviation
(RSD) of 14 % (Table 3) for all data excluding the engine idle condition (23% N1).
This provides high confidence for the use of the TAP for in-flight or field



measurements of aircraft-engine nvPM mass emissions, when filter-loading
corrections (Section 3.3.3) are correctly applied.
The PSAP, on the other hand, showed much greater variability, with an RSD of 36%
(Table 3). This is substantially higher than the variability reported by a laboratory
intercomparison of PSAP and CAPS $PM_{SSA}$ (Perim De Faria et al., 2021) (that study
did not report a statistic comparable to RSD). Although the PSAP has been
observed to deviate up to a factor of two higher in cases of high organic aerosol
loading or reduced filter transmission (Lack et al., 2013), our data are restricted to
transmissions above 0.8. The fact that the PSAP shows great variability rather than
a fixed offset indicates that the issue is not due to a systematic error such as an
inaccurate MAC or flow rate calibration.   We note that the TAP and PSAP were
operated with reduced sample flow rates of 0.05 L min$^{-1}$ and 0.1 L min$^{-1}$,
respectively, (5 to 10% of nominal settings) to extend the life of their filter media
while sampling the high soot concentrations in the aircraft exhaust.  Under these
conditions, detector noise and small fluctuations in sample flow have a magnified
effect on resulting derived absorption coefficients. We suspect that the
measurements would have been significantly more precise if the instruments had
been operated at nominal flows, although this would have required changing
filters after each test point. Consistent with our hypothesis, we note that
Nakayama et al. (2010) observed substantially larger variability in PSAP
measurements at 0.3 than at 0.7 standard litres per minute. We also note that Bond
et al. (1999) did not observe an impact of flow rate when changing from 1 to 2
litres per minute.
Figure 10 plots as a function of particle GMD the same relative TAP and PSAP $EI_m$
data shown in Figure 9b. No clear trend of this ratio with size is evident, although
the measurements become somewhat more scattered at smaller sizes for the SAF1
data set, where signal to noise is lower (GMD and $EI_m$ were correlated, see the
below discussion of Figure 12). Figure 10b includes the size-dependent PSAP
correction function reported by Nakayama et al. (2010) (their Equation 8), with
shading representing a 1σ uncertainty. Those authors predicted the true
absorption values using Mie theory for nigrosin particles of diameter 100 to
600 nm and refractive index 1.685−0.285$i$. Thus, their correction factor is





conceptually equivalent to our $EI_m$/mean-$EI_m$. Extrapolating their correction
function down from 100 nm to 15 nm gives values ranging from 4 to 8, whereas
our measurements are close to 1.0. This discrepancy may be attributed primarily
to the extrapolation, and possibly also to the fact that we have measured solid
nvPM particles rather than liquid nigrosin. Overall, it is clear that the variability in
our PSAP data is not sufficiently predicted by the GMD.
Overall, our data show that any possible size dependency in the TAP and PSAP
response is smaller than the observed variability between samples. The TAP and
PSAP data exhibit relative standard deviations (RSD) of 19% and 16%,
respectively, for samples with GMD > 25 nm. Future studies may consider
correcting PSAP and TAP measurements by the ratios shown in Table 3, which
represent the ratio between the calibrated aerosol-phase nvPM mass
measurements and the previously uncalibrated PSAP and TAP measurements, for
data above 25 mg $kg_{fuel}^{-1}$ and N1 > 40%.
4.5    Instrument performance for fuels with different composition
Figure 11 shows a category plot of the ratio $EI_m$/mean-$EI_m$ (that is, the ordinate of
Figure 9) for the different instruments. Data below 100 mg / $kg_{fuel}$ have been
excluded as this ratio reflects only noise in that region (Figure 9). The symbols
have been sized by mean N1. The data have been coded by symbol and colour to
reflect the 6 fuels used in this study, although JP-8 measurements are few in
number due to the $EI_m$ of the data set (CFM56-2C1 with JP-8) being typically below
25 mg / $kg_{fuel}$.
Figure 11 shows that no substantial difference can be seen for these instruments
for the nvPM EIm for fuels with different composition; the spread in the data for a
given fuel is larger than the difference between fuels. Outliers tend to be associated
with low N1 (small symbols). Because low N1 corresponds to both lower
concentrations (lower signal-to-noise) and lower exhaust velocities relative to
ambient wind speeds, these outliers are not surprising.
The instruments in Figure 11 show a linear response to nvPM mass and operate on
a range of physical principles. This observation indicates that no instrument was



uniquely sensitive to changes in particle size over the observed range, since $EI_m$
was correlated with GMD (Figure 12), as is typical of aviation engines (Saffaripour
et al., 2020). We note that the response of all of these instruments is proportional
to the MAC of the sample, so that it remains possible that the sample MAC changed
with GMD or $EI_m$.
4.6    Influence of LII laser fluence
An additional experiment was performed to test the hypothesis that the laser
fluence of the LII 300 may not be sufficiently high to heat nvPM to incandescence
in aircraft-engine PM emissions from SAFs at different engine thrusts. This
hypothesis is related to electron microscopy evidence (Vander Wal et al., 2014)
showing that the degree of graphitization of aircraft-engine soot may be
substantially lowered at low thrusts. A lower degree of graphitization may result in
a lower LII signal if the 1064 nm MAC is lower (resulting in a lower maximum
temperature being reached) or if part of the laser energy leads to carbon annealing
rather than thermal excitation (Botero et al., 2021; Ugarte, 1992; Vander Wal and
Choi, 1999). If correct, this hypothesis would mean that the nvPM concentrations
reported by an LII 300 operated at reduced fluence would be lower than those of a
reference LII 300. Higher fluences are also required for nvPM internally mixed
with volatile PM, as some laser energy may be lost to volatile evaporation
(Michelsen et al., 2015).
Figure 13a illustrates the experiment we performed to test this hypothesis. The
figure presents data for SAF1 only; results for other fuels were similar. One
"reduced-fluence" LII 300 was programmed to change its Q-switch delay from
140 μs to 240 μs, with a randomized order. In this experiment, lower Q-switch
delays corresponded to higher laser fluence; the lowest Q-switch delay was the
optimal one for this system. Another "reference" LII 300 operated with no change
to its Q-switch delay. Figure 13a shows that the reduced-fluence LII reported lower
mass concentrations when its Q-switch delay was increased, but returned to the
expected values when its Q-switch delay was reduced.
We defined $R_{LII}$ as the ratio of nvPM mass concentrations reported by the reduced-



fluence and reference LII 300 instruments. Figure 13b shows that $R_{\mathrm{LII}}$ was a
function of Q-switch delay, and therefore laser fluence, for all engine thrust
conditions. This observation is expected, since LII signals are lower at lower
fluence (Michelsen et al., 2015) and since we calculated $R_{\mathrm{LII}}$ without taking this
effect into account. We have verified in our laboratory that Q-switch delay is
inversely proportional to laser fluence for this system and that saturation effects
are negligible.
A trend of decreasing $R_{\mathrm{LII}}$ with decreasing N1 is evident at moderate and low Q-
switch delays, which can be interpreted as indicating that the nvPM was more
graphitic at higher N1 conditions. However, $R_{\mathrm{LII}}$ reached a plateau at high fluence
(smaller Q-switch delay), which is the region where the LII 300 normally operates.
This plateau was reached at all engine thrusts, with a broader range for the plateau
at higher thrusts and a decreasing range as the thrust was lowered. Therefore, the
LII 300 has sufficient fluence and can be expected to perform well for SAF blends
at all engine thrust conditions.
## 5    Conclusions
For multiple instruments measuring nvPM number, size, and mass, we observed
no evidence of anomalous instrument responses to the exhaust emissions
produced by SAF blends relative to reference fuels (REFs) combustion in an IAE
V2527-A5 engine. The GMD, GSD, and $EI_{num}$ data for all fuels fell within 20%, 5%,
and a factor of 2 of their mean, respectively. Anomalous instrumental responses
would have resulted in two groups of data for these parameters, which was not
observed. However, a difference between $EI_{num}$ for instruments located on
different-length sampling lines was noted and attributed to a greater sensitivity of
$EI_{num}$ than $EI_m$ to the penetration function.

The majority of nvPM mass measurements by the real-time instruments (CAPS
$PM_{SSA}$, LII 300, MSS+, PAX) agreed to within 30% of their geometric mean
(reference mean), for $EI_m$ above 100 mg/ $kg_{fuel}$. This lower limit corresponded to a
mass concentration of approximately 10 µg m$^{-3}$ (the conversion of $EI_m$ to mass
varies because the emitted $[CO_2]$ varies), which was the noise level of these





instruments. The ratio of each real-time measurement with the reference mean
was close to unity (maximally 1.24, minimally 0.78) and indicated good precision
(all RSDs ≤ 17%).

Integrative nvPM $EI_m$, calculated from PSD measurements or filter attenuation
(TAP and PSAP), fell within a factor of two of the reference mean. The ratio of each
integrative measurement with the reference mean was further from unity
(maximally 1.50, minimally 0.88) and variability was higher precision (all RSDs
≤ 36%). The variability in TAP data was notably low at 14%, and the variability in
PSAP data was notably high at 36%, likely due to its operation at a reduced flow
rate.

Two other instrument- and fuel composition-specific observations were made. A
dedicated experiment showed that changing the laser fluence of an LII 300 could
influence its reported nvPM mass concentrations at low to moderate fluences. By
maintaining sufficiently high fluence a plateau region was established, irrespective
of thrust or fuel, where reported nvPM mass concentrations were stable and not
influenced by experimental conditions. Second, additional measurements of
emissions from JP-8 fuel combusted in a CFM56-2C1 engine indicated the presence
of very high concentrations of volatile nucleation-mode particles with
diameter < 20 nm. These measurements reflect a different engine, as well as a fuel
with a factor 20 higher sulfur content, and the increased total PM number
concentration is most likely attributable to the sulfur.

Overall, this study found that real-time instruments for the measurement of nvPM
emissions in aviation turbine engines are comparable whether conventional fuels
or SAFs are used. Since all real-time measurements were influenced by the MAC
and no independent measurement of nvPM mass was made, no conclusions about
the variability thereof can be made from this study.



### 5.1  Author contributions

BEA, PLC, TS, PL, GJS, PDW, and RML designed the study. JCC, TS, PLC, GJS, ECC, SA, PDW, RML, ZY, AF, MT, DS, WL, CR, PO, MS, and PL took the measurements. JCC, TS, BEA, RHM, MAS, ECC, SA, ZY analyzed the data with input from GJS, PL, RML, and AF. JCC prepared the figures. JCC and PL drafted the manuscript. All authors discussed the data interpretation and presentation.

### 5.2  Competing interests

RML and AF are employed by ARI, which produces the CAPS $PM_{SSA}$ commercially. ZY was employed by ARI at the time of the study.

### 5.3  Acknowledgements

We acknowledge the efforts of the flight and ground crews of the DLR ATRA, the NASA DC8, and the U.S. Air Force 86th Air Wing. We thank the ground staff of Ramstein Air Base for their operational support during this experiment.

### 5.4  Financial support

This research received funding from the Transport Canada project "TC Aviation — nvPM from renewable and conventional fuels". The campaign was funded by the DLR aeronautics program in the framework of the project "Emission and Climate Impact of Alternative Fuels (ECLIF)". MS&T and ARI received support from the U.S. Federal Aviation Administration (FAA) through the Aviation Sustainability Center (ASCENT) – a U.S. FAA-NASA- U.S. DoD-Transport Canada- U.S. EPA sponsored Center of Excellence for Alternative Jet Fuels and Environment under Grant No. 13-C-AJFE-MST, Amendment 010. A.F. was supported by funds from ARI. ATRA operational and fuel costs along with DLR scientists' participation was funded by the DLR aeronautics program in the framework of the project "Emission and Climate Impact of Alternative Fuels (ECLIF)". The U.S. FAA Office of Environment and Energy and the National Aeronautics and Space Administration Aeronautics Research Mission Directorate supported field and DC-8 operations, and participation of the U.S. researchers in the project.



## 5.5 Data availability

The data presented in Figures 4 to 10 are available in the Zenodo repository at

https://sandbox.zenodo.org/record/950512 as a spreadsheet file. Other data are

available from the authors upon request.

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





1060    7    Figures and Tables

1061    Table 1. Properties of the fuels used for the ground-based measurements (fuel

1062    samples acquired from wing-tank after test).

| Property | Method | JP-8 | REF3 | REF4 | SAF1 | SAF2 | SAF3 |
|---|---|---|---|---|---|---|---|
| Aromatics [vol%] | ASTM D1319 | 19.9 | 18.6 | 16.5 | 9.6 | 10.8 | 15.2 |
| Hydrogen H [mass%] | ASTM D7171 | 13.86 | 13.65 | 14.08 | 14.40 | 14.51 | 14.04 |
| Sulphur, total [ppm] | ISO 20884 | 1240 | 105 | 5.7 | 56.8 | 4.1 | 58.6 |
| Naphthalenes [mass%] | ASTM D1840 | 1.49 | 1.17 | 0.13 | 0.61 | 0.05 | 0.64 |
| Smoke point [mm] | ASTM D1322 | 23.0 | 23.0 | 27.0 | 30.0 | 30.0 | 28.0 |

1063

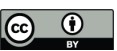


Table 2. Instruments used to measure nvPM and key measured properties. All instruments reported data at 1 second intervals except the SMPSs (45 second intervals for NRC and DLR, 60 seconds for NASA) and filter samplers. Instrument abbreviations are defined in the text.

| Operator | Instrument | Acronym | Species measured | Sampling duration [s] | Units | Penetration functions[d] |
|---|---|---|---|---|---|---|
| NASA | Particle soot absorption photometer | PSAP | nvPM[a] mass | 1 | µg m-3 | 1 |
| | Tricolor absorption photometer | TAP | nvPM[a] mass | 1 | µg m-3 | 1 |
| | Scanning mobility particle sizer | SMPS | Total PSD (10 to 278 nm) | 45 | particles cm-3, and µg m-3 | 1 |
| | Thermo-denuder with SMPS | TD-SMPS | nvPM PSD[b] (10 to 278 nm) | 45 | µg m-3 | 1, 4 |
| NRC | $CO_2$ sensor | LI-COR 7000 | $CO_2$ | 1 | ppmv | - |
| | Cavity-attenuated phase shift PM$_{SSA}$ monitor (660 nm) | CAPS (NRC) | nvPM[a] mass | 1 | µg m-3 | 1 |
| | Photoacoustic extinctiometer | PAX | nvPM[a] mass | 1 | µg m-3 | 1 |
| | Laser-induced-incandescence | LII 300 (NRC; 2x) | nvPM[b] mass | 1 | µg m-3 | 1 |
| | Catalytic stripper SMPS | CS-SMPS | nvPM PSD (8.6 to 278 nm) | 45 | particles cm-3 | 1, 3 |
| MST (NARS) | AVL Particle Counter Advanced | APC | nvPM number | 1 | particles cm-3 | 1, 2 |
| | Micro Soot Sensor | MSS Plus | nvPM[a] mass | 1 | µg m-3 | 1, 2 |
| | Laser-induced-incandescence | LII-300 (NARS) | nvPM[c] mass | 1 | µg m-3 | 1, 2 |
| | $CO_2$ sensor | LI-COR 840A | $CO_2$ | 1 | ppm | - |
| | Differential mobility spectrometer | DMS500 | Total PSD (5 to 1000 nm) | 1 | particles cm-3 | 1, 2 |
| ARI | Cavity-attenuated phase shift PM$_{SSA}$ monitor (630 nm) | CAPS (ARI) | nvPM[a] mass | 1 | µg m-3 | 1, 2 |

[a]nvPM measured via particulate absorption as equivalent BC (eBC). [b]Particle size distribution, here measured with respect to mobility diameter. [c]nvPM measured via laser-induced incandescence as refractory BC (rBC). [d]Numbers are indices corresponding to the penetration functions shown in Figure 4.





Table 3. Summary of the ratios between the $EI_m$ of individual instruments and the geometric mean of the Group 1 (real time) instruments. The corresponding raw data are shown in Figure 11. Regression: linear regression against Group 1 geometric mean weighted by standard deviations, with k = 2 uncertainties from fit. SD: standard deviation. RSD: Relative SD. Group 1: real time instruments. Group 2: integrative instruments.

| | | $EI_{mass}$ Ratio vs. Group 1 | | | Regression vs. Group 1 | |
|---|---|---|---|---|---|---|
| Group | Instrument | Mean | SD | RSD [%] | Intercept | Slope |
| 1 | CAPS PM$_{SSA}$ (ARI) | 0.84 | 0.08 | 10 | 12 ± 19 | 0.8 ± 0.1 |
| 1 | CAPS PM$_{SSA}$ (NRC) | 0.99 | 0.09 | 9 | -0.3 ± 0.8 | 1.01 ± 0.04 |
| 1 | LII (NARS) | 1.24 | 0.18 | 15 | 27 ± 6 | 1.03 ± 0.04 |
| 1 | LII (NRC-0331) | 1.07 | 0.1 | 9 | -15 ± 42 | 1.17 ± 0.16 |
| 1 | LII (NRC-0574) | 0.78 | 0.08 | 10 | -17.1 ± 2 | 0.88 ± 0.08 |
| 1 | MSS+ | 1.07 | 0.14 | 13 | 17.8 ± 5 | 0.92 ± 0.04 |
| 1 | PAX | 1.06 | 0.18 | 17 | -15 ± 1 | 1.21 ± 0.02 |
| 2 | CS-SMPS | 1.50 | 0.27 | 18 | 12 ± 22 | 1.02 ± 0.12 |
| 2 | TD-SMPS | 1.14 | 0.26 | 23 | -5 ± 1 | 1.47 ± 0.04 |
| 2 | PSAP | 0.89 | 0.32 | 36 | 8 ± 16 | 0.82 ± 0.08 |
| 2 | TAP | 0.88 | 0.12 | 14 | 6 ± 6 | 0.75 ± 0.02 |

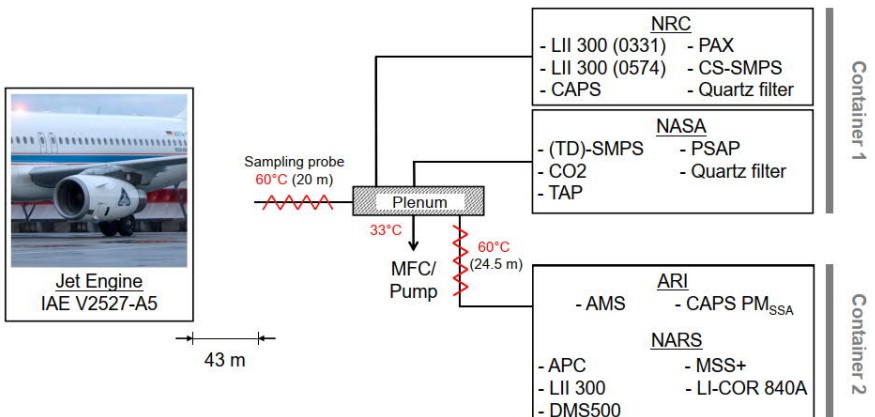

Figure 1. Schematic of sampling configuration behind the DLR ATRA aircraft. The length and flow rate of sampling lines from the manifold to the various instruments varied as described in the text. The NRC and NASA instruments were all placed within Container 1, while the NARS and ARI instruments were placed in Container 2. For simplicity, the figure omits a short heated line connecting the first plenum to the NARS. The ARI instruments were downstream of all NARS instruments except the DMS500 (see Lobo et al., 2016 for detailed NARS diagram). NARS = North American Reference System.

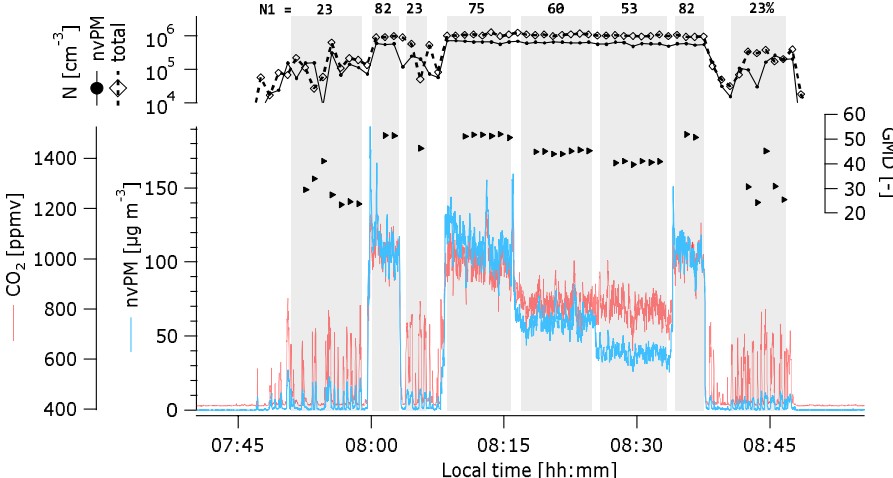

Figure 2. Illustration of a typical test run. Variation in the $CO_2$ concentration was not due to instrument noise, as illustrated by the $CO_2$ measurements prior to and following sampling. A representative nvPM mass instrument is shown by the blue trace. Sizing information (GMD) is shown by the black symbols (triangles: GMD; diamonds with dashed line: total PM number; spheres with solid line: nvPM number measured with the CS-SMPS).





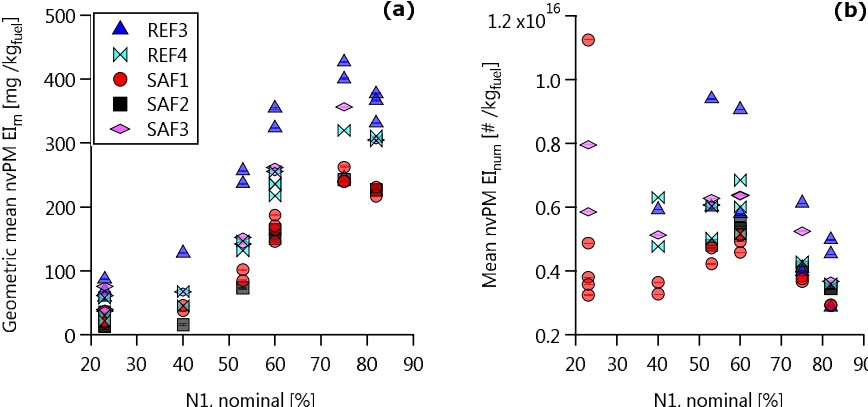

Figure 3. Relationship between nvPM (a) $EI_m$ and (b) $EI_{num}$ with N1 for all data obtained with the V2527-A5 engine. The trends shown in this plot are discussed further in the companion article (Schripp and NDMAX-Team, 2021). The ordinate values are the geometric mean discussed in the text.

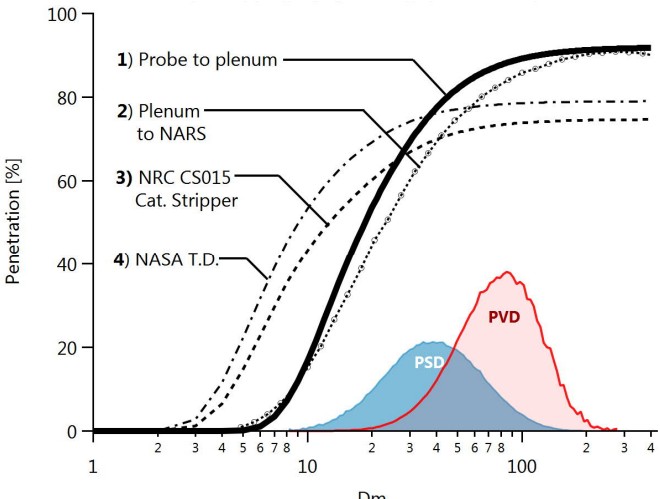

Figure 4. Penetration functions for the main probe-to-plenum sampling line as well as other components in the sampling system. Shaded areas illustrate a representative particle size (PSD) and volume (PVD) distribution measurement with GMD 34 nm and GSD 1.72. PSD data for all test points and instruments are provided in the supplement. NARS: North American Reference System; CS015: Catalytic Stripper; T.D.: thermodenuder.

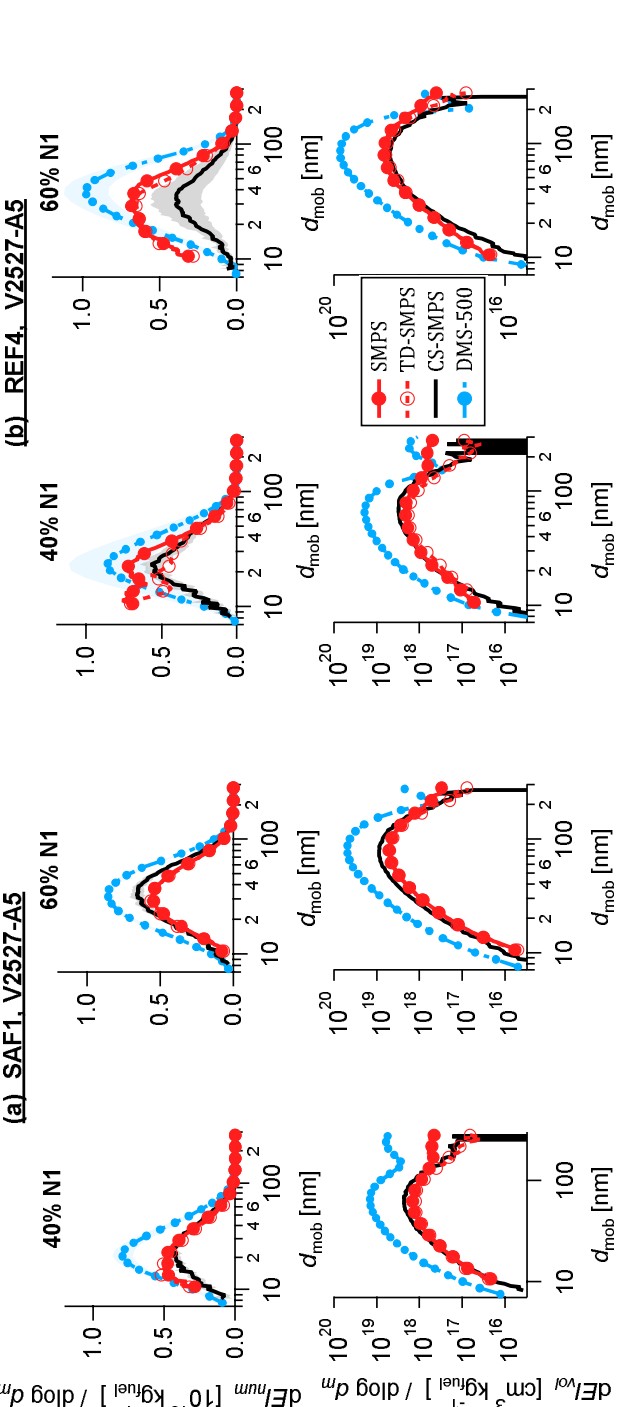

Figure 5. Selected PSDs illustrating the V2527-A5 engine with (a) SAF1 fuel and (b) REF4 fuel. Each panel shows 60% N1 on the right and a lower N1 on the left: 40% for (a), 60% for (b). Note that the TD-SMPS and CS-SMPS (red open circles and black line) represent nvPM, while the SMPS and DMS500 represent vPM.



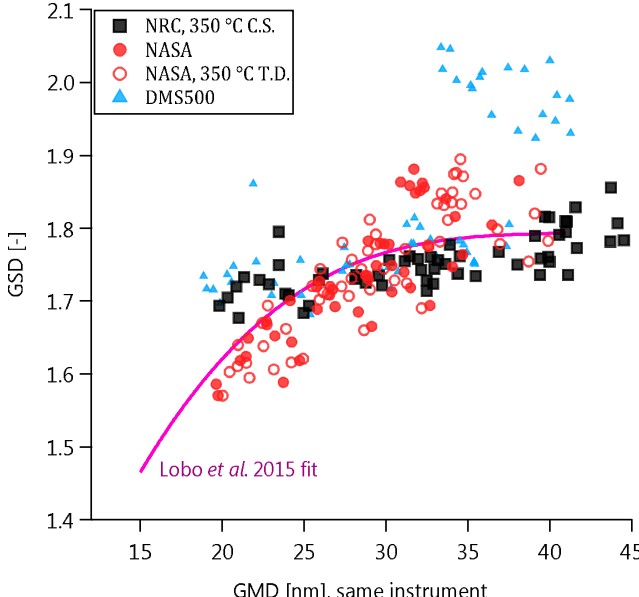

Figure 6. GSD versus GMD data as by measured by each particle sizer for all test points. Higher GSDs for the DMS500 correspond to bimodal PSDs (non-volatile and volatile modes). Note that size-dependent particle losses (see penetration functions in Figure 4) may affect both GSD and GMD. Based on Figure 12, the TD-SMPS (NASA) data may be more accurate than the CS-SMPS data (see text). Fit is from Lobo et al. (2015c).

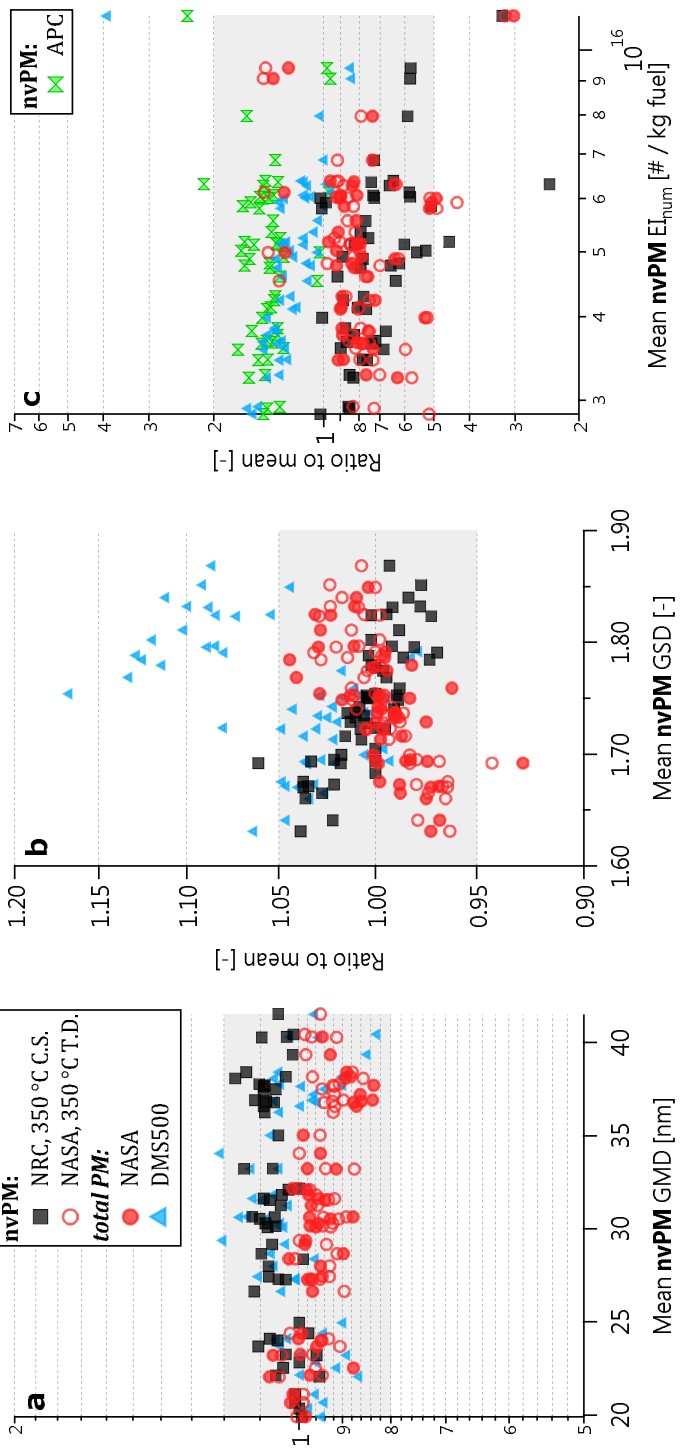

Figure 7. Comparison of size and number measurements in terms of GMD, GSD, and EI$_n$. Grey shading shows 20%, 5%, and 200% in GMD, GSD, and EI$_n$, respectively. In panels (a) and (b), mean is defined from the CS-SMPS (NRC) and TD-SMPS (NASA) data. In panel (c), the mean additionally includes the APC (NARS) data (the APC is in the NARS and uses a TSI 3790E CPC).



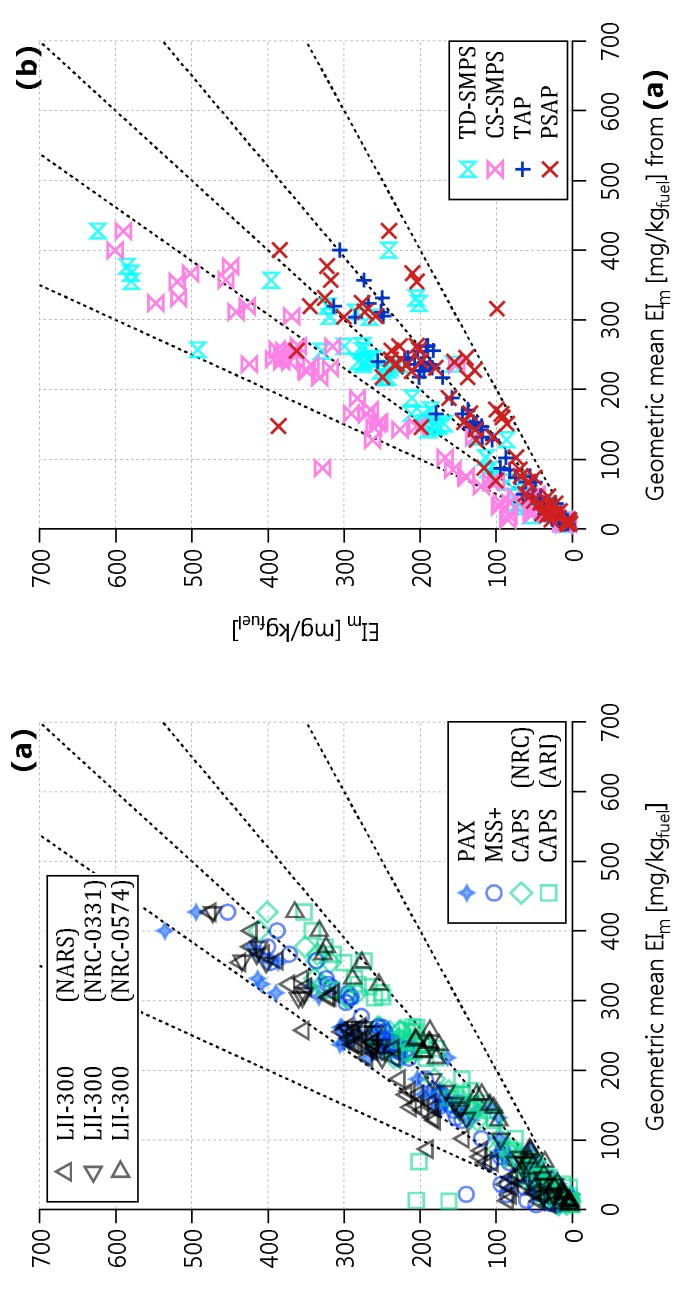

Figure 8. $EI_m$ scatterplot for (a) real-time and (b) integrative $EI_m$ measurements. The term integrative refers to SMPS measurements (mass concentrations estimated by assuming unit-density spheres) and filter photometer measurements (mass concentrations estimated using standard empirical relationships between light attenuation and light absorption). The abscissa of both panels is the geometric mean of all available data from the 7 real-time sampling instruments plotted in (a). Angled lines illustrate slopes of 2, $2^{-1}$, 1.3, $1.3^{-1}$, and 1.0.


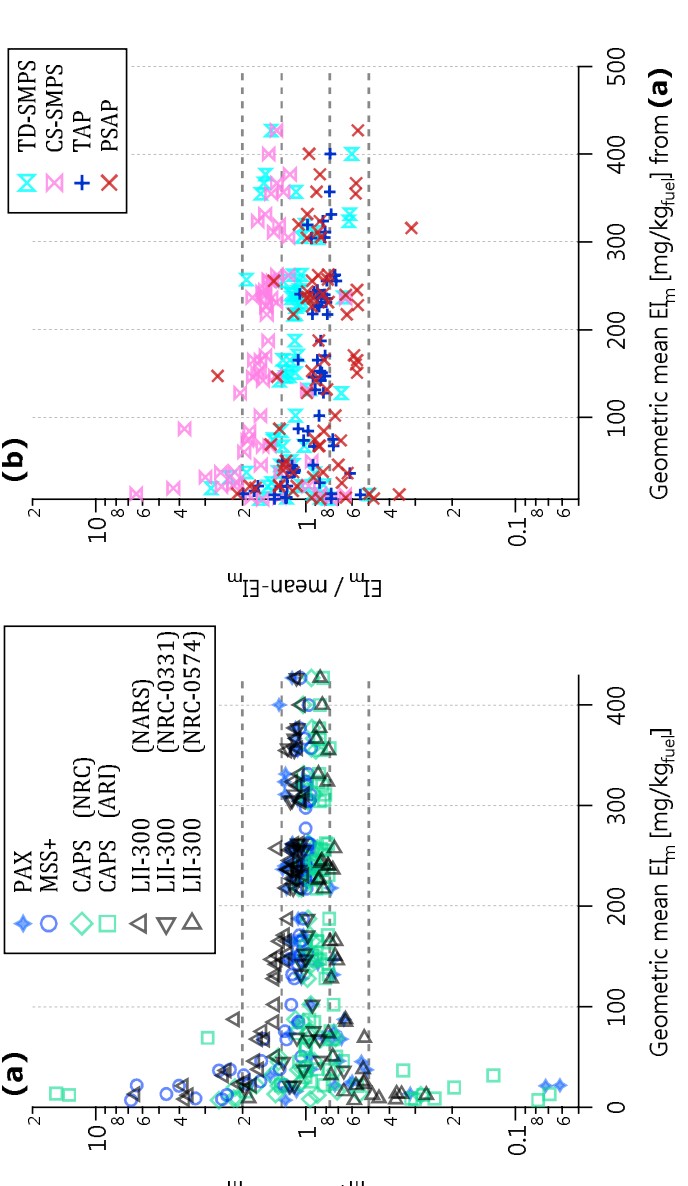

Figure 9. Ratio plots corresponding to Figure 8. The inner and outer horizontal lines show ratios of 2, $2^{-1}$, 1.3, $1.3^{-1}$, and 1.0. Agreement between the instruments is poorer at $EI_m < 100$ mg/kg$_{fuel}$, which corresponds to an approximate concentration of 10 μg m$^{-3}$ (the exact conversion factor varies with $CO_2$ concentration and fuel properties) and close to the limit of detection for most instruments.


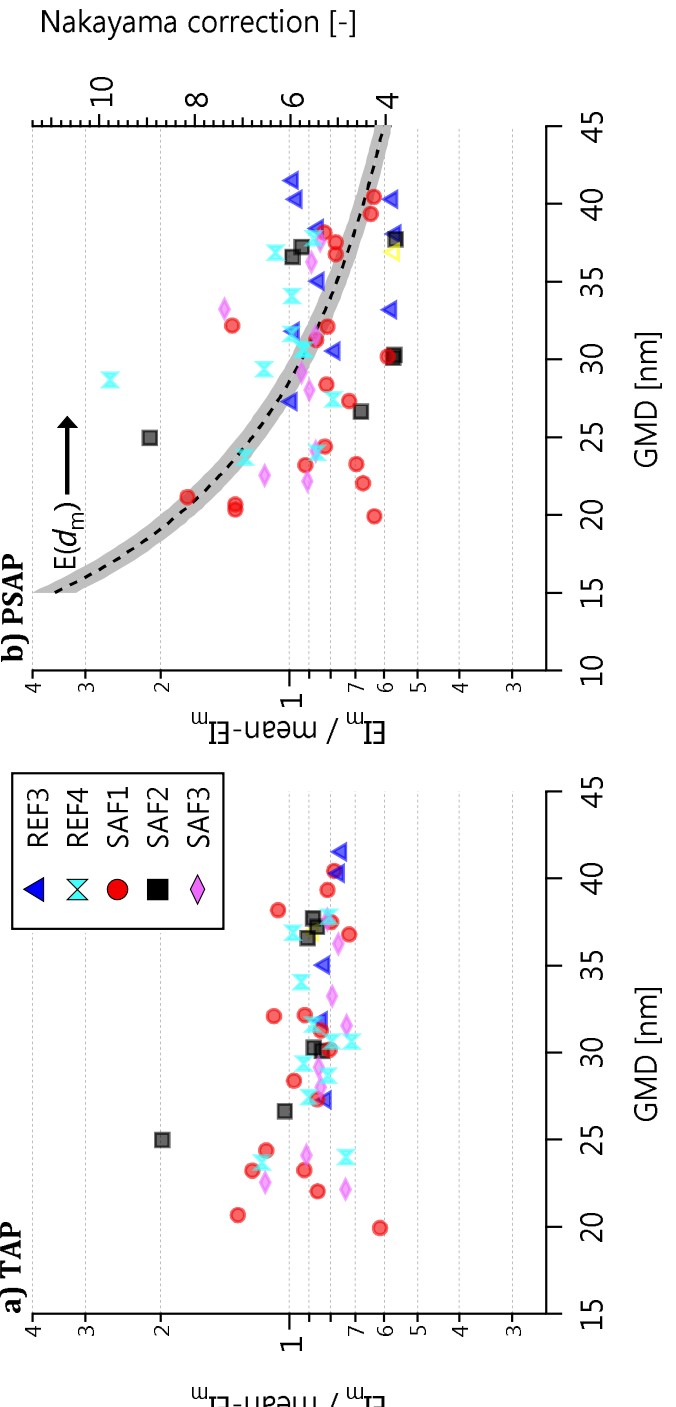

Figure 10. $EI_m$/ mean-$EI_m$ ratios from Figure 9 for the TAP and PSAP (the filter-based photometers) only, plotted as a function of geometric mean mobility diameter (GMD) to highlight potential size-dependent sensitivities of these instruments. The curve labelled $E(d_m)$ in b) plots the size-dependent PSAP correction factor given by Nakayama et al. (2010; Eq. 8) with 1σ uncertainties shaded.

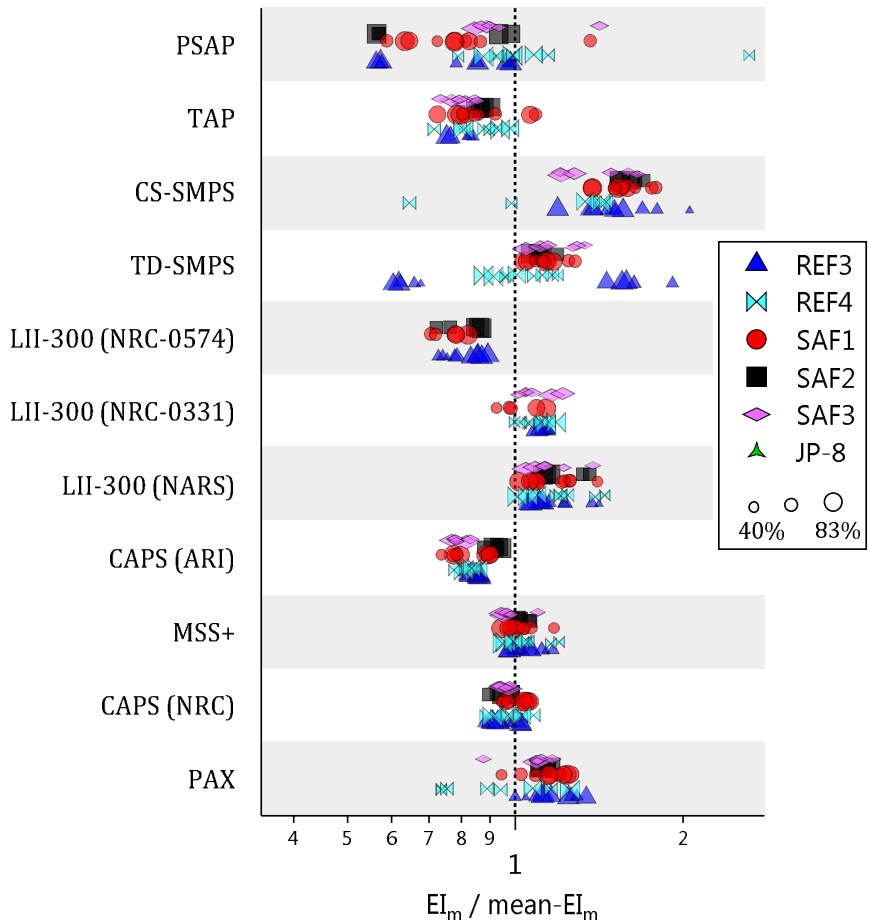

1133

Figure 11. Ratios of Figure 9 grouped by fuel. All fuels except JP-8 were combusted in the

V2527-A5 engine; JP-8 was combusted in the CFM56-2C1 engine. Shading is to guide the

eye. Symbols are sized by N1 thrust. Plot excludes data where $EI_m < 25$ mg/kg$_{fuel}$ and N1

thrust below 40% to minimize the effects of instrument noise and wind speed,

respectively, on the ratios.



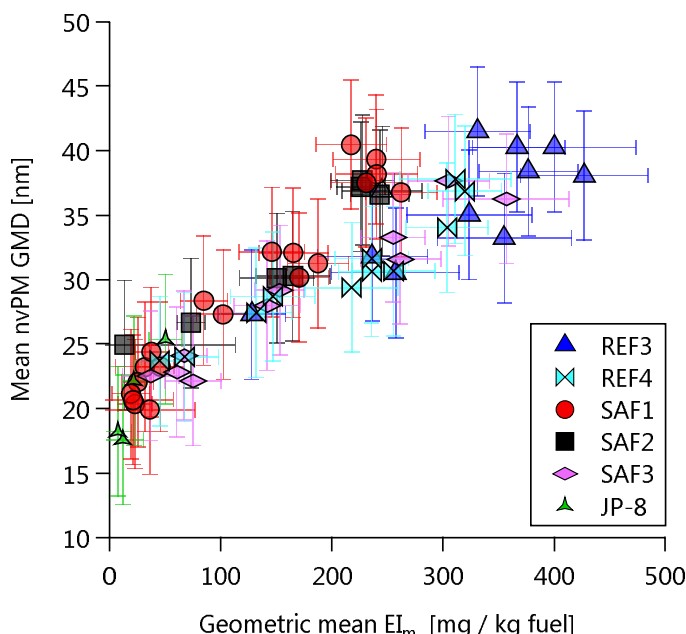

Figure 12. Scatterplot of the mean nvPM GMD within test points against geometric mean nvPM $EI_m$ from Figure 8a. The correlation with GMD and $EI_m$ indicates that Figure 9 implicitly represented different particle sizes.



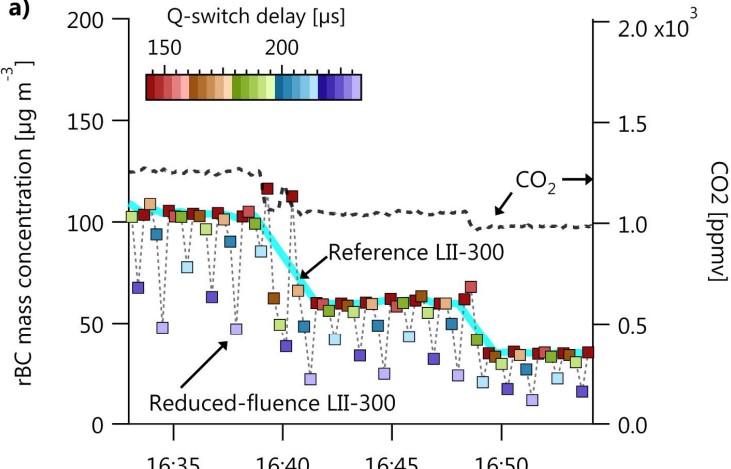

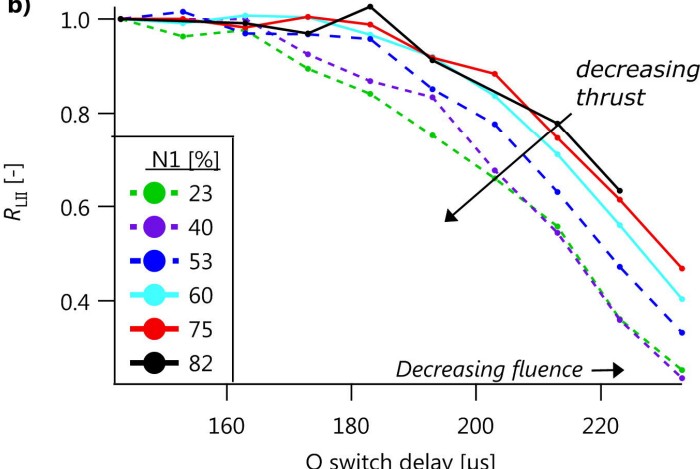

Figure 13. (a) LII 300 experiment time series, where one LII 300 was operated with increased Q-switch delays to reduce its laser fluence (squares) and the other was operated at standard fluence (solid line). $CO_2$ data are also shown for context. (b) The ratio $R_{LII}$ of the concentration reported by the reduced-fluence LII divided by the reference LII. It is evident from (b) that the standard high-fluence conditions generate data that are independent of N1 thrust, and that moderate- and low-fluence conditions (Q-switch delays greater than about 165 to 185 μs) display a weak dependence on thrust.