# Peer review of "Aircraft-engine particulate matter emissions from conventional"

_Atmospheric Measurement Techniques, 2021_

## Referee Comment (RC1)

**General review:**

This manuscript compares different measuring techniques for aircraft gas-turbine-emitted nvPM mass number and size, and total number and size from exhaust sampled at 43m of a V2527-A5 and a CFM56-2C1 aircraft engine burning a range of sustainable and conventional aviation fuels as part of the ECLIF 2 test campaign.

The manuscript is well written, and the data presented is novel and relevant to the scientific community. The data processing is of good quality; however, I found that the interpretation of the consistency between nvPM number and mass emission indices was not entirely addressed and sometimes misleading, particularly for mass and size. Please note that I wasn't able to access the supplementary information and table S1 which may answer some of my comments.

**Major comments:**

- **Loss correction:**
  - Penetration function 1 measurement: Would you be able to add the size distribution characteristics of the nebulised ammonium sulfate (it could only be in the SI)? Was it representative and the size distributions you typically measured during the test campaign (i.e., GMD ~20-40 nm)?

  - Penetration function 2: I am unsure of what you mean here. Do you mean you used the UTRC model to predict size-dependent losses in this section (as can be seen in Figure 4) or did you use the full N/M method that outputs a correction factor for nvPM number and mass? If you only used the UTRC model, which particle size distribution did you use? If you used the N/M method, how did you correct for losses to the DMS-500? The loss correction methodology you used would affect the interpretation of your results, and therefore it should be clearly explained.

- **Figure 5:**
  - 40% N1 with REF4 graph: It appears the SMPS+TD also measures part of a peak < 10 nm, which appears to be volatile given the SMPS+CS doesn't see it. Does that mean the TD is not 100% efficient at removing volatiles? This should be discussed. Also in line 49, you discuss that an nvPM mode < 10 nm was observed with the CFM56 engine. Can you confirm that it was a nvPM mode and not a volatile mode?

  - On multiple PSDs, the start of a large size mode can be observed (~200 nm) which could indicate you were measuring shed particles (unless it came from the engine) or oil. Were you performing regular cleanliness checks? Did you use in-line cyclones to all your analysers? If this was shedding, it could significantly impact nvPM mass measurements < 10 ug/m3 and would affect EIm estimation from integrated particle size measurement. This is not discussed anywhere in the manuscript; hence discussion regarding this should be added.

- **Particle size statistics; GMD and GSD**: It is not clear whether you compared the measured particle size distributions or if they were corrected for particle loss to a common sampling

point (plenum or probe) using a bin-by-bin approach with measured PSD and the penetration functions (or another method?). Please clarify this in the main text and in the figure titles.

- **Section 4.3: Consistency between number-based emission indices of nvPM and vPM**

  o APC Vs SMPS number: Have you considered that both SMPSs were just under-reporting due to the large corrections performed within the SMPS software (losses in the DMA, the poor charging efficiency of the bipolar charger, non-linearity of the CPC response)? When were the SMPSs last serviced and calibrated? Were the SMPSs compared with the APC on the same source prior to the test campaign? As currently written, this section implies that SMPSs are more precise at measuring nvPM number than the standard regulatory compliant APC. I don't agree with your conclusion that the APC was likely overcorrected only because you found the two SMPSs to agree with each-other. What if there was two APC in agreement and only one SMPS?

  o L560: The DMS-500 was measuring unstripped aerosol and therefore could be picking up volatiles in comparison with the other nvPM EInum analysers, which could also explain why it was reporting higher values.

  o Were all the size/number analysers within the recommended 12 months service and calibration period? If not, that could explain some of the disparities observed between the different size analysers. For example, the DMS-500 is calibrated for number and size to a traceable standard but drifts over time and Cambustion only certifies measurement precision of 10% for size and 20% for number within 12-months. I suggest you add discussion on the calibration uncertainty associated with all analysers.

  o Figure 7: It is biased that you only used the SMPSs to calculate the mean and then you compared that mean to the DMS GMD and GSD given it wasn't included in the mean calculation. Why didn't you apply the same methodology as for the nvPM mass analyser, calculating the mean using all the different analysers?

- **Section 4.4: Consistency between mass-based emission indices**

  o Scatter < 100 mg/kg fuel: Are you sure this reflects the noise levels of the instruments? Analysers like the LII and MSS are, to my knowledge, capable of precisely measuring down to 1 ug/m3. Is the difference between the LIIs bias or scatter? Did you consider shedding from your system could impact nvPM mass measurement as mentioned in a comment above or that potentially inaccurate calibration caused this difference (see comment below)? Please discuss this in the manuscript.

  o I would expect the scatter between the three LIIs to be lower than reported given they are the same analyser, particularly the two NRC LIIs given they are next to each-other and presumably calibrated in the same manner. This is not addressed or discussed in the text, which is surprising given the detail that goes into the fluence sector. Could it have to do with the calibration performed for these analysers? Did the laboratory diffusion flame show ICAO annex16 applicability? If not, it could well explain some of the scatter you observe for EIm. This is something worth discussing in the manuscript.

○ SMPS based EIm: I find the interpretation of this section misleading, as it suggests the SMPS is nearly as good as an LII or MSS at measuring nvPM mass. First, the SMPSs generally do not capture the full VSD (as can be seen in figure 5). Secondly, you assumed unit density for the CS-SMPS but the particle density could well be below 1 g/cm3, particularly given particle effective density decreases with increasing size and the mass is carried by the larger particles. Using integrated particle size measurement to derive mass is strongly influenced by the density you select and should be discussed. Furthermore, I disagree with L613-L618; I believe the main reasons for the higher SMPS predictions is density assumption, the measurement uncertainty for the size bins >100 nm where the number count is very low, and potential shedding interference (see comments above). It is also surprising to me that in all figure 5b there is excellent agreement between the 2 stripped SMPS VSD's however it is then observed in figure 8b that there is no agreement. I am unsure how this is consistent if the same assumption regarding density is made?

○ L645: if the TAP and PSAP require a filter change at each test point to operate optimally, doesn't it make them not suitable for aircraft nvPM mass measurement? Particularly given the mass loading you've experienced were typically lower than certification measurements as you were sampling 43m downstream of the engine. If that's the case, then I suggest re-writing the abstract and conclusion to highlight this.

- **Conclusion**: I suggest re-writing the conclusion considering the comments above. For example, L574, I disagree that 10 ug/m3 is the noise level of the instruments. I suggest replacing "instrument" by "instrument calibration and sampling methodology".

**Minor comments:**

- General: The manuscript could do with more cross-referencing for the reader to find information more easily.

- L39 & L169: replace "sampling" by "measurement"

- L117: add "minimum" before 50%.

- L122: I don't believe it's true that the APC, MSS and LII are the only commercial instruments that satisfy the SARP. For example, a Dekati DEED and a Grimm or TSI CPC is a commercial system that satisfies the SARP. Please clarify.

- L152 & L183: I don't think you can reference something that hasn't been published yet or that that hasn't been peer reviewed yet. As a reviewer to this paper it is hard to critically appraise the statements and conclusions without being able to see the detailed experimental set-up and graphs addressing fuel effects etc.

- L219: Why was the plenum only maintained at 33°C? It seems odd to me that you first sampled via a 60°C heated line, then a 33°C plenum and then other 60°C heated lines. This would promote thermophoretic loss (although very small) and could cause water to condense. Please justify.

- L230: Why did you use a 25m line between the Dekati diluter and the NARS instruments? Was it because you couldn't get container 2 any closer to container 1? Wouldn't it have been better to use a shorter line to minimise diffusional losses and reduce your loss correction uncertainty which accounts for some of the discrepancies observed in your data? Given the 4:1 dilution ratio and sampling position the NARS system was not in compliance anyway hence could have been further optimised.

- L244: Please quantify what you mean by "good agreement".

- L255: The DMS-500 measures from 5 nm not 10 nm. Also, what is the size range of the EEPS?

- L317: drift not drifted.

- L371: Typo $CO_2$ not CO2

- L376: remove "from".

- L518: What inversion matrix was used for processing the DMS-500 data? Please add to manuscript.

- L550: It's not true the SMPS measures at 10 nm with a 100% efficiency as lots of corrections are applied (charging efficiency, loss through DMA and tubing). Please add "corrected" before 100%.

- Figure 7: The x-axis is labelled "mean nvPM xxx", however some total PM is also shown (as labelled in the legend). Please clarify and correct x-axis.

- L783: therefore not therefor

---

## Author Comment (AC1)

Author responses to reviews of
amt-2021-320:
"Aircraft-engine particulate matter emissions from conventional
and sustainable aviation fuel combustion: comparison of
measurement techniques for mass, number, and size"
by J. C. Corbin et al.

**1. RC2**

I struggle with the assessment of this work: while the all the methods and results
presented are of respectable scientific quality, I think there is a lack of focus in terms of
relevance and scope for AMT. There is no novelty in concepts or data treatment and it is
not clear what the real scientific value of the study is. For regulatory purposes there is
little value due to the non-compliant sampling system, non- existent pre experiment
calibration etc. The scientific value is also limited – I understand the argument for
connecting ground measurements to cruise at altitude data, but for that purpose, a more
focused effort with a better experimental design that would allow tracking down
sampling/ conditioning from instrument issues would be beneficial. With the current
manuscript one gets the impression that it is a side product of a bigger effort and was not
carefully thought through when the experiment was conducted – which is not necessarily
a problem if the reader does not get this impression, but I currently do.

Our third-last paragraph in the introduction provided some
justification. This paragraph was followed by a misplaced paragraph
describing the measurement campaign – that misplaced paragraph has now
been moved to Methods, and a new sentence added to the third-last
paragraph. The full paragraph is now (new text underlined):

The standardized system components are not easily adaptable for use
on aircraft for measurement of cruise level nvPM emissions.  Consequently,
there are no comparable in-flight engine-emissions data available for
developing and validating models that predict cruise nvPM-emissions based
on engine certification data.  Particle size distribution measurements are also

not included in the standardized system, which are important for assessing the effects of fuels, operating conditions, and engine technologies on the environmental impacts of PM emissions. Thus to advance our understanding of aircraft engine emissions and the factors that control them as well as to develop a large and consistent observational data base, it is important to evaluate the relative performance of other diagnostic instruments that are not prescribed in the standardized protocol but meet these needs. Such instruments must be evaluated for their response to nvPM and total PM emissions from aircraft engines using standardized and non-standardized systems, and for measurements at the engine exit plane and downstream of the engine in the near field, since these instruments are typically used with minimal change to their operating parameters for a wide range of sampling conditions. Very limited data are available in the literature for this purpose, and no data have yet been published for SAFs.

Thus, this manuscript features one aspect of the detailed analysis that is one facet of a large collaborative project. The manuscript, with its analysis of the response of instruments to variations in the properties of the particulate emissions with fuel type, has implications for in-flight measurements of SAF emission factors, standardized vs. non-standardized measurements, and total vs. non-volatile PM emissions.

Major comments:

The comparison of the mass measurement is somewhat biased experimentally (due to distance to the engine, dilution, detection limits and long lines etc.) to higher thrust levels. At these thrust levels it is not a major surprise that there is not much variability in instrument responses (little OC, larger aggregate sizes, soot properties less influenced by fuel type etc.). I also tend to disagree with the authors conclusion that a 30-50% difference is a "comparable" especially for the near real time in situ instruments such as MSS LII and CAPS. Would be good to point this out to the reader, or even split the discussion for cruising relevant (i.e. 50-70% thrust) and near idle thrusts this might improve the lack of

relevance pointed out above.

The bias to "higher" thrust levels is only caused by the rejection of some test points at 23% thrust. Some 23% data was retained, and the remainder of the data spans 40% to 83%. This range of thrusts is substantial.

We agree with the reviewer's "disagreement" that 30-50% is not really "comparable". We did not intend to imply that a 30-50% disagreement is not statistically significant. We believe that it is significant and implies a systematic bias (e.g. calibration drift or imperfect line-loss corrections) between the instruments. The reviewer may have the impression that we believed otherwise because our discussion focussed on the larger disagreements of the SMPS and filter-based instruments (up to a factor of 2).

When we searched the manuscript for the word "comparable" we could not find that word used to imply no statistical significance. We do agree that we made that implication by omission. We modified Section 4.4.1:

The agreement of the real-time measurements to within 30 % is . larger than the calibration uncertainties of the individual instruments, and suggests an influence of systematic biases (e.g. in instrument calibration or penetration corrections). There is no evidence of systematic differences between absorption and LII measurements, which might have been hypothesized if coatings of volatile PM on the light-absorbing nvPM had enhanced absorption.

Here we also added the underlined sentence to introduce a new hypothesis about why the measurements might differ.

We have not observed any systematic differences by thrust. Figure 11 shows this for N1 thrusts from 40% to 83%. Any differences between

instruments are larger than differences between thrusts. So, we have not taken the reviewer's last suggestion.

It would beneficial to show the comparison of measured concentration as a function of CO2 ( at least in the SI)

All requested information was provided in the supplementary data file. The measured $CO_2$ increment ranged from 0 ppm to 929 ppm, with median 384 ppm.

We take this comment to be related to the comparison of the mass instruments, for example in Figures 8 and 9. We agree that the relevant axis for a mass instrument comparison is mass concentration. However, the instruments shown in Figures 8 and 9 were located on different sampling lines and experienced different levels of dilution. Therefore, we were forced to compare these instruments in terms of EIm rather than mass concentration.

SMPS EIm derivation: this work makes the impression that an SMPS measures the volume size distribution with high precision and there is furthermore no need to apply a size dependent effective density (which I believe is crucial for larger sizes). It would be beneficial for the discussion to elaborate on this based on previous experiences on helicopter or jet engines [...]

The reviewer is correct that we omitted a description of the SMPS PSD mass integration in our Methods section. We now added the following paragraph:

Finally, the SMPS PSDs were converted to equivalent mass concentrations by the integrated PSD approach, described in detail by Momenimovahed and Olfert (2015). In brief, the equivalent mass of each SMPS-reported mobility diameter was calculated using an effective density of 1000 kg m$^{-3}$, which has been shown to produce better than 20% accuracy relative to more complete, size-resolved effective densities (Durdina et al., 2014).

---

## Author Comment (AC2)

Author responses to reviews of

 amt-2021-320:

"Aircraft-engine particulate matter emissions from conventional and sustainable aviation fuel combustion: comparison of measurement techniques for mass, number, and size"

   by J. C. Corbin et al.

2. RC1

General review:

This manuscript compares different measuring techniques for aircraft gas- turbine-emitted nvPM mass number and size, and total number and size from exhaust sampled at 43m of a V2527-A5 and a CFM56- 2C1 aircraft engine burning a range of sustainable and conventional aviation fuels as part of the ECLIF 2 test campaign. The manuscript is well written, and the data presented is novel and relevant to the scientific community. The data processing is of good quality; however, I found that the interpretation of the consistency between nvPM number and mass emission indices was not entirely addressed and sometimes misleading, particularly for mass and size. Please note that I wasn't able to access the supplementary information and table S1 which may answer some of my comments.

*We thank the reviewer for their time and comments.*

*According to an emailed communication with AMT, our references to*

*the "Supplement" should have been references to a "Data Availability"*

*section, and the reviewer was emailed this information as well.*

Major comments:

- Loss correction:

o Penetration function 1 measurement: Would you be able to add the size distribution characteristics of the nebulised ammonium sulfate (it could only be in the SI)? Was it representative and the size distributions you typically measured during the test campaign (i.e., GMD ~20-40 nm)?

We added statements of the GMD and GSD to the text:

The ratio of the NASA to NRC PSDs (GMD 30 nm, GSD 1.7) then provided a first estimate of the penetration function.

We note that it is not essential that the GMD be similar to those measured during the campaign, since the penetration function is size-resolved. It is only essential that sufficient number counts are measured in each bin to obtain reasonable uncertainties.

o Penetration function 2: I am unsure of what you mean here. Do you mean you used the UTRC model to predict size-dependent losses in this section (as can be seen in Figure 4) or did you use the full N/M method that outputs a correction factor for nvPM number and mass? If you only used the UTRC model, which particle size distribution did you use? If you used the N/M method, how did you correct for losses to the DMS-500? The loss correction methodology you used would affect the interpretation of your results, and therefore it should be clearly explained.

The full paragraph starts with Particles may be lost to the walls of sampling lines or to deposits on those walls. The fraction of particles penetrating a given system varies with size, according to a characteristic penetration function. Four penetration functions were applied in this study … (Figure 4), to clarify that we refer to size-dependent functions as seen in Figure 4. We clarified the subsequent sentence by changing

Function 2 was calculated using the standard loss calculation methodologies provided in SAE documents AIR6504 (SAE, 2017) and ARP6481 (SAE, 2019).

To

Function 2 was calculated using the standard equations for line penetration, as detailed in the loss calculation methodologies provided in SAE documents AIR6504

(SAE, 2017) and ARP6481 (SAE, 2019).

Here the text specifies that the calculations produced Function 2 as shown in Figure 4, which is size-resolved. The following new paragraph was added to state this explicitly:

All reported data are corrected for these penetration functions. Size-resolved data (SMPS) were corrected using the size-resolved penetration functions shown in

Figure 4. Size-integrated data (all other instruments) were corrected by weighting the penetration functions by the corresponding measured SMPS PVDs. The correction factors are given in the Data Availability section.

- Figure 5:

o 40% N1 with REF4 graph: It appears the SMPS+TD also measures part of a peak < 10 nm, which appears to be volatile given the SMPS+CS doesn't see it.

Does that mean the TD is not 100% efficient at removing volatiles? This should be discussed. Also in line 49, you discuss that an nvPM mode < 10 nm was observed with the CFM56 engine. Can you confirm that it was a nvPM mode and not a volatile mode?

Our data do not allow us to identify whether these small particles were non- volatile or represent an imperfect performance of the CS and TD.

We added the above statement after the description of the <10nm mode. We kept this brief to avoid speculation.

o On multiple PSDs, the start of a large size mode can be observed (~200

nm) which could indicate you were measuring shed particles (unless it came from the engine) or oil. Were you performing regular cleanliness checks? Did you use in-line cyclones to all your analysers? If this was shedding, it could significantly impact nvPM mass measurements < 10 ug/m3 and would affect EIm estimation from integrated particle size measurement. This is not discussed anywhere in the manuscript; hence discussion regarding this should be added.

There is some evidence for an increase in SMPS-calculated volume at larger particle sizes in Figure 5a, at both 40% and 60% N1. If these large particles indicated the presence of a large aerosol mode which varied independently from the primary mode (e.g. if they were emitted by some other process than the engine itself), they would introduce a $EI_m$-dependent bias in the ratio of SMPS-based $EI_m$ to other instruments, which was not observed (Section 4.4.2).

We added the text above to the Results. We did not mention shedding explicitly as we feel that a mention requires a citation of a study proving its importance. Shedding is extremely unlikely in our study; our main sampling line was brand new and was baked prior to analysis. There was no evidence of shedding in zero and background air measurements. Also, the large particle mode in Figure 5 represents volatile particles, and is very likely related to oil. Our AMS data indicated the presence of oil-related mass fragments. However, the AMS data are out of scope of the present study.

Additional arguments can be put forward as follows, that we feel are excessive for the manuscript:

Figure 2 shows that the PM mass and number concentrations were close to zero (number is off-scale due to the log scale) for background conditions. Filtered-inlet conditions would therefore be even lower. There is no evidence of shedding here.

Figure 9 shows that there is no change in the ratio of EIm/mean-EIm at lower EIm. Therefore, the bias between SMPS and mass-based instruments was not a function of EIm. If shedding contributed to line concentrations, then its contribution would be larger at lower EIm, and the SMPS would be biased lower at lower EIm (because it would not see all of the shed particles).

- Particle size statistics; GMD and GSD: It is not clear whether you compared the measured particle size distributions or if they were corrected for particle loss to a common sampling point (plenum or probe) using a bin-by-bin approach with measured PSD and the penetration functions (or another method?). Please clarify this in the main text and in the figure titles.

We added a paragraph to Methods:

All reported data are corrected for these penetration functions. Size- resolved data (SMPS) were corrected using the size-resolved penetration functions shown in Figure 4. Size-integrated data (all other instruments) were corrected by weighting the penetration functions by the corresponding measured SMPS PVDs. The correction factors are given in the Data

Availability section.

And a comment in Results:

Figure 5 shows selected PSDs […] The PSDs are corrected for line penetration as described above

And modified Figure 4's caption:

These functions have been used to correct all other presented data.

- Section 4.3: Consistency between number-based emission indices of nvPM and vPM

o APC Vs SMPS number: Have you considered that both SMPSs were just under- reporting due to the large corrections performed within the SMPS software (losses in the DMA, the poor charging efficiency of the bipolar charger, non-linearity of the

CPC response)? When were the SMPSs last serviced and calibrated? Were the

SMPSs compared with the APC on the same source prior to the test campaign? As currently written, this section implies that SMPSs are more precise at measuring nvPM number than the standard regulatory compliant APC. I don't agree with your conclusion that the APC was likely overcorrected only because you found the two SMPSs to agree with each-other. What if there was two APC in agreement and only one SMPS?

Here, the Reviewer has helpfully included detailed questions to justify their valid criticism. However, we believe that this criticism results from a miscommunication and not a difference of scientific interpretation. We did not intend to imply that we believed the SMPSs to be more reliable than the APC, nor to use the SMPSs as reference to diagnose problems with the APC.

The old paragraph is:

Figure 7c compares the measured vPM and nvPM $EI_{num}$ with the mean nvPM $EI_{num}$ (i.e., mean of the NRC CS-SMPS, NASA TD-SMPS, and NARS APC. The grey shading shows that all instruments agreed to within a factor of 2. The APC and DMS500 nvPM $EI_{num}$ were both typically higher than the two similar SMPSs. Substantial variability between the two SMPSs was also observed.

In Figure 7c, the penetration-corrected APC $EI_{num}$ are approximately 50% larger than the SMPS $EI_{num}$ under all conditions. Our measured PSDs rule out the possibility that 50% of particles were not seen by the SMPS. Therefore, we attribute the difference between APC and SMPS results to uncertainties in the APC or SMPS penetration correction functions (Figure 4), i.e., we hypothesize that this difference would not have been observed had the instruments all sampled from the same plenum from comparable sampling lines.

The rewritten paragraph is:

In Figure 7c, the penetration-corrected APC $EI_{num}$ are approximately 50% larger than the SMPS $EI_{num}$ under all conditions. Our measured PSDs rule out the possibility that 50% of particles were not seen by the SMPS. Therefore, we attribute the difference between APC and SMPS results to uncertainties in the APC or SMPS penetration correction functions (Figure 4), i.e., we hypothesize that this difference would not have been observed had the instruments all sampled from the same plenum from comparable sampling lines.

We have not specifically addressed the Reviewer's technical comments about SMPS uncertainty because our revisions to the manuscript already cover these details.

Running all counting instruments on the same source, with equivalent lines, would have be an excellent experiment to perform. However, due to the practical limitations of working from separate containers at a field site with limited access, we were not able to perform this experiment.

Note that the only other relevant text in the manuscript is in the abstract and also does not imply a preference for the SMPS data:

[abstract] The commercial instruments used included TSI SMPSs, a Cambustion DMS500, and an AVL APC, and the data also fell within approximately 50 % of their geometric mean.

o L560: The DMS-500 was measuring unstripped aerosol and therefore could be picking up volatiles in comparison with the other nvPM EInum analysers, which could also explain why it was reporting higher values.

We agree and had made this statement. Now, we have further clarified (see previous point of response).

o Were all the size/number analysers within the recommended 12 months service and calibration period? If not, that could explain some of the disparities observed between the different size analysers. For example, the DMS-500 is calibrated for number and size to a traceable standard but drifts over time and Cambustion only certifies measurement precision of 10% for size and 20% for number within 12-months. I suggest you add discussion on the calibration uncertainty associated with all analysers.

Please see two responses above. All instruments for used for measurement of particulates in this manuscript have larger uncertainties than experienced with measuring gas phase properties, for instance. It is not unusual for 20% uncertainty with these instruments. This is well known in aerosol science (Kulkarni, Willeke, and Baron, 2011), and adding a discussion on this topic to this manuscript would not represent a contribution to the literature.

Kulkarni, P., Baron, P. A., & Willeke, K. (2011). Aerosol measurement: principles, techniques, and applications. John Wiley & Sons.

o Figure 7: It is biased that you only used the SMPSs to calculate the mean and then you compared that mean to the DMS GMD and GSD given it wasn't included in the mean calculation. Why didn't you apply the same methodology as for the nvPM mass analyser, calculating the mean using all the different analysers?

This is a misunderstanding. We used all available data. Figure 7c's caption states

"In panel (a) and (b), mean is defined from the CS-SMPS (NRC) and TD-SMPS (NASA) data. In panel (c), the mean additionally includes the APC (NARS) data"

- Section 4.4: Consistency between mass-based emission indices o Scatter < 100 mg/kg fuel: Are you sure this reflects the noise levels of the instruments? Analysers like the LII and MSS are, to my knowledge, capable of precisely measuring down to 1 ug/m3. Is the difference between the LIIs bias or scatter? Did you consider shedding from your system could impact nvPM mass measurement as mentioned in a comment above or that potentially inaccurate calibration caused this difference (see comment below)? Please discuss this in the manuscript.

The reviewer's theoretical noise level for LII and MSS is correct, but in practice LIIs sometimes have higher detection limits (unpublished data from NRC) and MSSs suffer from background-gas absorption as does the CAPS (Elser et al., 2019).

We addressed shedding in our response above. Shedding is one of multiple potential reasons for the scatter, but we have no evidence for it in our work. In fact, we have evidence against it (above). Discussing only this hypothesis would suggest to the reader that it is more likely than the alternatives; we wish to avoid that implication.

o I would expect the scatter between the three LIIs to be lower than reported given they are the same analyser, particularly the two NRC LIIs given they are next to each-other and presumably calibrated in the same manner. This is not addressed or discussed in the text, which is surprising given the detail that goes into the fluence sector. Could it have to do with the calibration performed for these analysers? Did the laboratory diffusion flame show ICAO annex16 applicability? If not, it could well explain some of the scatter you observe for EIm. This is something worth discussing in the manuscript.

Here the reviewer has correctly focussed on the scatter between the three LIIs, rather than the ratio between them. One reason why the apparent scatter is higher is that the two NRC LIIs (0331 and 0574) were not operated continuously for the entire campaign. Consequently, there are only 5 points in Figures 7 and 8 where the two NRC LIIs can be directly compared with one another. Figure 13a shows that the scatter between these two LIIs is minimal over short time periods (compare the thick teal line with the dark red squares). Therefore, calibration is not a likely cause of the scatter. We speculate that some of the scatter between LIIs is also due to the penetration corrections, which were applied on a point-by-point basis and which were discussed in Section 4.3.

To minimize speculation, and because we have no direct evidence for penetration corrections causing the differences between LIIs, we hesitate to discuss this one hypothesis in detail. We consider it very likely that other unknown hypotheses may play a role.

o SMPS based EIm: I find the interpretation of this section misleading, as it suggests the SMPS is nearly as good as an LII or MSS at measuring nvPM mass. First, the SMPSs generally do not capture the full VSD (as can be seen in figure 5).

This is true, but:

Since the spread of nvPM $EI_m$ reported by the two SMPS systems was smaller than the bias, their difference relative to the reference $EI_m$ cannot be attributed to measurement biases (such as the limited size range detected by the instruments).

(underlined text is new)

Secondly, you assumed unit density for the CS-SMPS but the particle density could well be below 1 g/cm3, particularly given particle effective density decreases with increasing size and the mass is carried by the larger particles. Using integrated particle size measurement to derive mass is strongly influenced by the density you select and should be discussed.

It is generally true that the effective density function has a strong influenced on SMPS-based mass estimates. However, for the specific size and effective density functions expected for aviation soot, larger uncertainties may arise due to line penetration corrections than effective density. This has been shown nicely by Durdina et al. (2014) and cited on line 470 (Methods). The reason is that the line penetration correction for the small sizes of aircraft soot particles tends to be very large, relative to other soot sources. This point does bear repeating in Results, so we added:

(We reiterate that our assumption of constant effective density is expected to introduce negligible uncertainty for the small soot particles emitted by aircraft turbine engines; Durdina et al., 2014).

Furthermore, I disagree with L613-L618; I believe the main reasons for the higher SMPS predictions is density assumption, the measurement uncertainty for the size bins >100 nm where the number count is very low, and potential shedding interference (see comments above). It is also surprising to me that in all figure 5b there is excellent agreement between the 2 stripped SMPS VSD's however it is then observed in figure 8b that there is no agreement. I am unsure how this is consistent if the same assumption regarding density is made?

In response to the previous comment, we pointed out the systematic study by Durdina et al. (2014) which proves that the density assumption incurs negligible error.

The reviewer also hypothesizes that low number counts at large size bins caused the SMPS errors. This would imply a uniquely higher scatter in the SMPS data at small concentrations (since the upper edge of the distribution, which contains most of the mass, would become 'small' first), which is not observed in Figure 9b.

To the reviewer's comment that agreement between the SMPS PVDs in Figure 5b appears better than Figure 8b, we point out that there are a few experiments where the SMPS data overlap in 8b. We did manually choose the example in Figure 5b and undoubtedly introduced some bias, but all SMPS data have been presented using summary statistics in other figures and the raw data for which are included in the Data Availability section. Also, the validity of the summary statistics such as GMD and GSD in Figure 7, which are more appropriate for lognormal distributions, was checked for all presented data.

o L645: if the TAP and PSAP require a filter change at each test point to operate optimally, doesn't it make them not suitable for aircraft nvPM mass measurement? Particularly given the mass loading you've experienced were typically lower than certification measurements as you were sampling 43m downstream of the engine. If that's the case, then I suggest re-writing the abstract and conclusion to highlight this.

Indeed, this is the least convenient aspect of the TAP and PSAP. But these instruments are tiny (can be operated handheld) and the sacrifice is necessary for aircraft measurements.

We made no relevant statements in the Conclusions, and the only relevant Abstract statement is:

The commercial instruments used were one TAP, one PSAP, and two SMPSs. These techniques are used in specific applications, such as on-board research aircraft to determine PM emissions at cruise.

It remains true that these instruments "are used" and, since the abstract is already very long, we have decided not to modify the sentence.

- Conclusion: I suggest re-writing the conclusion considering the comments above. For example, L574, I disagree that 10 ug/m3 is the noise level of the instruments. I suggest replacing "instrument" by "instrument calibration and sampling methodology".

We changed to:

the noise level of these instruments in our sampling setup

We cannot ascribe the observations to calibration issues, which would instead cause systematic bias. However, the calibrations could be due to background interferences, which become more important at low concentrations.

Minor comments: - General: The manuscript could do with more cross-referencing for the reader to find information more easily.

> We have added a couple of cross-references in the process of this review.

- L39 & L169: replace "sampling" by "measurement"

> Done

- L117: add "minimum" before 50%.

> Fixed

- L122: I don't believe it's true that the APC, MSS and LII are the only commercial instruments that satisfy the SARP. For example, a Dekati DEED and a Grimm or TSI CPC is a commercial system that satisfies the SARP. Please clarify.

> The reviewer is correct. Rather than attempt to complete our list of commercial instruments, we have removed this statement, to allow for future instruments which may enter the market.

- L152 & L183: I don't think you can reference something that hasn't been published yet or that that hasn't been peer reviewed yet. As a reviewer to this paper it is hard to critically appraise the statements and conclusions without being able to see the detailed experimental set-up and graphs addressing fuel effects etc.

> This is a fair comment, but the two papers are "companion" papers. We should have provided a draft manuscript. At the reviewer's request we are happy to provide a copy of the companion paper, which is now submitted to Fuel and cited as such (Schripp et al.)

- L219: Why was the plenum only maintained at 33°C? It seems odd to me that you first sampled via a 60°C heated line, then a 33°C plenum and then other 60°C heated lines. This would promote thermophoretic loss (although very small) and could cause water to condense. Please justify.

This was a practical limitation and will be corrected in future work.

- L230: Why did you use a 25m line between the Dekati diluter and the NARS instruments? Was it because you couldn't get container 2 any closer to container 1? Wouldn't it have been better to use a shorter line to minimise diffusional losses and reduce your loss correction uncertainty which accounts for some of the discrepancies observed in your data? Given the 4:1 dilution ratio and sampling position the NARS system was not in compliance anyway hence could have been further optimised.

This was done because the goal of the experiments was to have the NARS use its standardized sampling line. The penetration function of this sampling line has been characterized in detail.

- L244: Please quantify what you mean by "good agreement".

This statement was used to justify the selection of one instrument over another:

"The $CO_2$ measurements from the NASA LI-COR 7000 were in good agreement with those taken by DLR [...] but had a faster response time and were therefore used as the reference for instruments in Container 1."

We acknowledge that "good agreement" is not a scientific assessment, but given that the response times differed, the data would need to be deconvoluted before any point-by-point statistics are used. The difference in response time was related to a difference in flow rates. Since we have therefore not sought to systematically compare these two instruments we have avoided any quantitative statements here.

- L255: The DMS-500 measures from 5 nm not 10 nm. Also, what is the size range of the EEPS?

    DMS500: fixed
    EEPS: 5.6 to 560 nm. Added.

- L317: drift not drifted.

    fixed

- L371: Typo $CO_2$ not CO2

    fixed

- L376: remove "from".

    fixed

- L518: What inversion matrix was used for processing the DMS-500 data? Please add to manuscript.

    Log-normal inversion with a bimodal calibration matrix. Added.

- L550: It's not true the SMPS measures at 10 nm with a 100% efficiency as lots of corrections are applied (charging efficiency, loss through DMA and tubing). Please add "corrected" before 100%.

    Text changed (see above)

- Figure 7: The x-axis is labelled "mean nvPM xxx", however some total PM is also
shown (as labelled in the legend). Please clarify and correct x-axis.
The total PM is not included in the X axis. The X axis is correct.
- L783: therefore not therefor
Thereof not therefore. Unchanged. Thank you for the comments.

3. Other changes
We also made the following changes to the manuscript.
1. We realized that the reported aromatic concentrations for SAF1
and SAF2 in Table 1 were incorrect. They are now fixed.
2. In Section 4.1.1 we clarified that only the 'ordinate data' and not
'the measurements' were normalized by the mean.
3. We removed the citation to the manuscript by Anderson et al. (in
prep.).

---

## Referee Report (RR1)

I'd like to thank the authors for their thorough explanations and the modifications they've made during this first round of reviewing. I'm generally satisfied with the answers and modifications to my original concerns. However, I still have a couple of concerns as listed below.

- **SMPS based EIm**: I disagree with your rebuttal about the "negligible" impact of particle effective density on EIm derived from SMPS size measurement (L410 & L747). You quote Durdina et al. 2014 paper stating an uncertainty of 20%, but this uncertainty is for $ksl_{mass}$ calculation in which the engine exit VSD is divided by the instrument-location VSD, and therefore in which the particle effective density has a much smaller impact than deriving total mass from a size measurement. Furthermore, Durdina et al. 2014 paper is only reporting data for one engine for which the measured particle effective density was near 1 g/cm3, when other aircraft engines have displayed particle effective densities ranging from 0.2 to 1.9 g/cm3 (Saffaripour et al. 2019 https://doi.org/10.1016/j.jaerosci.2019.105467). For example, if the particle effective density for the engine you investigated was to be ~0.5 g/cm3, then the SMPS-derived EI mass would be over-reported by a factor 2 (i.e., 100% uncertainty) with a particle effective density assumption of 1 g/cm3. My concern is that the SMPS was significantly under-reporting number (50% less than APC), but your "high" particle effective density assumption made the SMPS based EIm appear to agree better. I suggest re-phrasing to highlight that there are significant uncertainties associated with the use of an assumed particle effective density when deriving EI mass from a particle size distribution on an unknown engine which may not have a particle effective density ~1 g/cm3 at any given powers.

- **EI number loss correction factor**: In your summary data spreadsheet, the "*number line loss correction factor*" can be seen to fluctuate between 1.33 to 2.66. Given the sampling system length between the probe and the APC/DMS (container 2), I would expect much larger loss correction factors (e.g., ranging between 2 and 10). What does this "*number line loss correction factor*" correspond to? It should also be different for the different number analysers. Can you please clarify why the loss correction you've applied does not bring better closure for EI number? Can you please also clarify if APC number was corrected for VPR loss and CPC cut point.

- **Loss correction in general**: I'd like to see more details for the loss corrections as I still find it hard to work out how you've corrected each instrument for particle loss. L495, you added a statement where you wrote "*Size-integrated data (all other instruments) were corrected by weighting the penetration functions by the corresponding measured SMPS PVDs*". I assume you've only used the PVD to correct for the mass analysers and you used the PSD for the number analysers; Did you also use the PVD to correct for number analysers (APC, DMS, etc) (otherwise, your number loss correction factors will be wrong)? Also, have you tried using the DMS-500 data to correct for losses to the NA system (APC, MSS) instead of only using SMPS data? Also, where the additional losses in the SMPS TD & CS and APC VPR corrected for? Without a clear description of the loss correction, it's hard for me to assess whether the loss correction could be further improved towards better agreement.

- **Figure 8 (a)**: I'd like to see more discussion about Figure 8(a) to better understand one of your main findings which is that the EI mass agreement was generally within 30% of the geometric mean for real-time mass measurement > 100 mg/kg fuel. For example, the MSS data seems to mostly be on the 1:1 line, the CAPS data seem to mostly be below the mean, and the LII data seem to be randomly scattered. Why do you think you observe such differences?

---

## Author Response (AR2)

**Author responses to 2nd reviews of**
**amt-2021-320:**
**"Aircraft-engine particulate matter emissions from conventional and sustainable aviation fuel combustion: comparison of measurement techniques for mass, number, and size"**
**by J. C. Corbin et al.**

We are grateful to the Reviewers for their additional efforts in identifying further opportunities for improvements to our manuscript. We respond point-by-point in the following.

**1. Reviewer 3**

*Reviewer report on the manuscript "Aircraft-engine particulate matter emissions from conventional and sustainable aviation fuel combustion: comparison of measurement techniques for mass, number, and size" by Corbin et al.*

*General: this work compares the response of various PM mass and number instruments when sampling in the near field of a V2527-A5 and a CFM56- 2C1 aircraft engine. The manuscript is well written, and the data processing is of good quality; however, the content is of somewhat limited scientific relevance. The comparison between the real time and filter based mass measurements is of high relevance but is unfortunately hampered see below.*

*Major:*

*The drastically reduced flows in the filter based clap and psap are a major concern and make the shown comparison not very useful. While the authors point out the potentially added noise, there are also other problems associated with this e.g. particle losses within the instruments, more undefined spot size, potential dependence on commonly/normally occurring leaking in these type of instruments. The authors discuss this adequately by citing other literature, but it is very important to clearly mention that the instruments were not operated according to their specifications.*

Presumably the reviewer did not notice this in their first round of comments, so we added a statement to highlight this in the abstract:

*The TAP and PSAP were operated at 5% and 10% of their nominal flow rates, respectively, to extend the life of their filters.*

and conclusions:

*as noted, the TAP and PSAP were operated at 5% and 10% of their nominal flow rates, respectively*

and caption of Figure 10:

*Note that the TAP and PSAP were operated at 5% and 10% of their nominal flow rates, respectively, for all measurements in this study.*

And a footnote in Table 3:

*[a]PSAP operated at 10% of its nominal flow rate. [b]TAP operated at 5% of its nominal flow rate.*

We did not extend the discussion which the reviewer described as "adequately citing other literature".

*Minor:*

*Ln 28/29 for an easier read I would add "absolute" in front of magnitude of emissions*

Done

*Ln 44: albedo is albedo in plural (CAPS PMSSAs) therefore I would remove the s but would add "instruments" for an easier read*

Done

*Ln 47 to 49. The use of integrative seems to be a little misleading in general (different things are integrated in the two techniques). The main difference is for the filter measurements is that they are not performed in situ*

Internally, we discussed this at length during preparation of the manuscript. We had originally considered the terms "in situ" vs. "filter-based", but this was modified because "in situ" is sometimes used to mean

"analyzed on site" rather than "analyzed in the aerosol phase" (e.g. photometers versus TOA or solvent extraction and analysis). Ultimately we found that no one expression is perfect and we made sure to define our terms at first use.

*Ln 122/123 As in the PMP, the catalytic stripper is not the key requirement an evaporation tube could also be used as stripper. The key is the additional dilution step after the stripper to prevent re nucleation*
Changed to "volatile particle remover".
*Ln 128 There is also a more recent publication by Durdina Empa?*
Cited

*Ln 376: I do not think one can call them "low cost"… and "portable" as mentioned in the response to reviewers. They are long term monitoring devices with great sensitivity (10 -100x better than a LII) but with a limited temporal resolution. Theoretically, they should actually work really well for low SSA/ strongly absorbing aerosol as measured here if operated correctly, but only with low filter loadings…*
On line 376 (tracked changes version) we wrote low-cost and low-maintenance, rather than portable. Low-maintenance fits the reviewer's comment that they are long-term monitoring devices. Low-cost is an objective fact relative to other instruments in this study! Also, the instruments are indeed more lightweight than many others, which is an advantage in aircraft-based measurements.
We agree with the reviewer's theoretical expectations, which is why we included Figure 10 and the associated discussion.

*Ln 512: please clarify wording what has the turbine/ fan speed to do with thrust? Isn't the low-pressure turbine linked to the fan?*
We changed to "Nominal low-pressure jet-engine primary fan speeds" to minimize confusion.

*Ln 771 to 783: Valuable discussion see major comment above – please clearly mention that the instruments were not operated according to their specification*

Done – see above.

*Ln 802 to 806: Since the instruments were not operated according to their specs, this correction is definitely not valid and should not be done – please delete this paragraph.*

We kept the paragraph because it is cited in companion papers where the same instruments were used at the same flow rate, but we added a clear statement about the flow rate modification. We also added footnotes to Table 3 stating this flow rate modification.

*Ln 824 might be typical for SAC combustors but not for other engine technologies*

The cited study is a recent review that included double annular combustor technologies. We believe the statement is justified by the present literature.

*Ln 862 more crystalline structure (e.g. A. Liati ES&T or Papers from Vander Wal group),*

Cited both groups, thank you for pointing out the opportunity.

*Figure 8. Could this high variability in the integrated SMPS data also be explained by the fast scanning (45s) which pushes the limits of Scanning DMAs transfer functions theory etc...*

Considering that we measured a single engine with a single sampling system, we believe that the likelihood of variability due to issues with the SMPS inversion is minimal and does not justify a mention.

**2. Reviewer 2**

*I'd like to thank the authors for their thorough explanations and the modifications they've made during this first round of reviewing. I'm generally satisfied with the answers and modifications to my original concerns. However, I still have a couple of concerns as listed below.*

*SMPS based EIm: I disagree with your rebuttal about the "negligible" impact of particle effective density on EIm derived from SMPS size measurement (L410 & L747).*

*You quote Durdina et al. 2014 paper stating an uncertainty of 20%, but this uncertainty is for $ksl_{mass}$ calculation in which the engine exit VSD is divided by the instrument-location VSD, and therefore in which the particle effective density has a much smaller impact than deriving total mass from a size measurement. Furthermore, Durdina et al. 2014 paper is only reporting data for one engine for which the measured particle effective density was near 1 g/cm3, when other aircraft engines have displayed particle effective densities ranging from 0.2 to 1.9 g/cm3 (Saffaripour et al. 2019 https://doi.org/10.1016/j.jaerosci.2019.105467). For example, if the particle effective density for the engine you investigated was to be ~0.5 g/cm3, then the SMPS-derived EI mass would be over- reported by a factor 2 (i.e., 100% uncertainty) with a particle effective density assumption of 1 g/cm3. My concern is that the SMPS was significantly under-reporting number (50% less than APC), but your "high" particle effective density assumption made the SMPS based EIm appear to agree better. I suggest re-phrasing to highlight that there are significant uncertainties associated with the use of an assumed particle effective density when deriving EI mass from a particle size distribution on an unknown engine which may not have a particle effective density ~1 g/cm3 at any given powers.*

The reviewer's arguments are well-made and thorough. To incorporate these points, we modified the text as follows:

*The CS-SMPS data were systematically higher than the geometric mean, potentially due to an overcorrection of the penetration of large particles to the SMPS or due to uncertainty in the effective density that must be assumed when converting SMPS data to $EI_m$. As noted in Section 3.3.3, we assumed an effective density of 1000 kg m$^{-3}$ based on the work of Durdina et al. (2014). Considerable uncertainty could be introduced due to this assumption, as the effective density of the nvPM particles (Momenimovahed and Olfert, 2015) may vary with the monomer diameter (Abegglen et al., 2014; Durdina et al., 2014) and/or shape of soot aggregates.*

*EI number loss correction factor: In your summary data spreadsheet, the "number line loss correction factor" can be seen to fluctuate between 1.33 to 2.66. Given the sampling system length between the probe and the APC/DMS (container 2), I would expect much larger loss correction factors (e.g., ranging between 2 and 10). What does this "number line loss correction factor" correspond to? It should also be different for the different number analysers.*

The reviewer correctly noticed that we only described mass-based correction factors in the text. We now modified the last paragraph of Section 3.4.2 to:

*All reported data were corrected using penetration functions. Size-resolved data (SMPS PSDs) were corrected using the size-resolved penetration functions shown in Error! Reference source not found.. Size-integrated data were corrected using either number-based (for the APC) or mass-based (for all other instruments). The number-based line loss corrections were calculated as the ratio of the corrected to uncorrected PSDs. The mass-based corrections were calculated using the corresponding ratio of PVDs. Correction factors for each test point are given in the Data Availability section.*

Two members of our team independently verified these calculations and the reviewer's expectation of larger correction factors was not met. The values between 1.33 and 2.66 correspond to ordinate values of 75% and 38% in Figure 4. Comparing the abscissa of Figure 4 with the GMDs in Figure 6, these values are found to be consistent.

Thank you for catching this oversight.

While working on the above correction, we also noticed a minor omission in Section 3.4.2, and fixed it by adding the sentence:

*Function 2 was adapted slightly for each instrument in the NARS due to the relatively small additional losses in the sampling lines of each instrument.*

*Can you please clarify why the loss correction you've applied does not bring better closure for EI number?*

Of course, applying this loss correction did improve closure compared to the uncorrected data. But errors potentially remained. We believe this is because Penetration Function 2 was not included in the line loss correction measurement, because the NARS had its own long sampling line (line 435, 542, Figure 1).

*Can you please also clarify if APC number was corrected for VPR loss and CPC cut point.*

Consistent with NARS results reported elsewhere, the APC was not corrected for VPR losses. The APC was not corrected for CPC cut-point because the measured PSDs suggesting the particle counts below 10 nm was negligible. The same is true for the SMPSs.

We note that further corrections for losses in the APC VPR would only make the discrepancy between APC and SMPSs even larger.

We modified the text to:

*APC (AVL Inc., which contains a TSI Model 3790E CPC and a volatile particle remover), ...*